# Blazing the trails before beating the path: Sample-efficient Monte-Carlo planning

**Jean-Bastien Grill**  **Michal Valko**
SequeL team, INRIA Lille - Nord Europe, France
jean-bastien.grill@inria.fr  michal.valko@inria.fr

**Rémi Munos**
Google DeepMind, UK*
munos@google.com

## Abstract

You are a robot and you live in a Markov decision process (MDP) with a finite or an infinite number of transitions from state-action to next states. You got brains and so you *plan* before you act. Luckily, your roboparents equipped you with a generative model to do some *Monte-Carlo planning*. The world is waiting for you and you have no time to waste. You want your planning to be efficient. *Sample-efficient.* Indeed, you want to exploit the possible structure of the MDP by exploring only a subset of states reachable by following near-optimal policies. You want guarantees on sample complexity that depend on a measure of the quantity of near-optimal states. You want something, that is an extension of Monte-Carlo sampling (for estimating an expectation) to problems that alternate maximization (over actions) and expectation (over next states). But you do not want to `StOP` with exponential running time, you want something simple to implement and computationally efficient. You want it all and you want it now. You want `TrailBlazer`.

## 1 Introduction

We consider the problem of sampling-based planning in a Markov decision process (MDP) when a *generative model* (oracle) is available. This approach, also called Monte-Carlo planning or Monte-Carlo tree search (see e.g., [12]), has been popularized in the game of computer Go [7, 8, 15] and shown impressive performance in many other high dimensional control and game problems [4]. In the present paper, we provide a sample complexity analysis of a new algorithm called `TrailBlazer`.

Our assumption about the MDP is that we possess a generative model which can be called from any state-action pair to generate rewards and transition samples. Since making a call to this generative model has a cost, be it a numerical cost expressed in CPU time (in simulated environments) or a financial cost (in real domains), our goal is to *use* this *model* as *parsimoniously* as possible.

Following *dynamic programming* [2], planning can be reduced to an approximation of the (optimal) value function, defined as the maximum of the expected sum of discounted rewards: $\mathbb{E}\left[\sum_{t \geq 0} \gamma^t r_t\right]$, where $\gamma \in [0, 1)$ is a known *discount factor*. Indeed, if an $\varepsilon$-optimal approximation of the value function at any state-action pair is available, then the policy corresponding to selecting in each state the action with the highest approximated value will be $\mathcal{O}\left(\varepsilon / (1 - \gamma)\right)$-optimal [3].

Consequently, in this paper, we focus on a near-optimal approximation of the value function *for a single given state* (or state-action pair). In order to assess the performance of our algorithm we measure its *sample complexity* defined as the number of oracle calls, given that we guarantee its *consistency*, i.e., that with probability at least $1 - \delta$, `TrailBlazer` returns an $\varepsilon$-approximation of the value function as required by the probably approximately correct (PAC) framework.

We use a *tree representation* to represent the set of states that are reachable from any initial state. This tree alternates maximum (`MAX`) nodes (corresponding to actions) and average (`AVG`) nodes (corresponding to the random transition to next states). We assume the number $K$ of actions is finite. However, the number $N$ of possible next states is either *finite* or *infinite* (which may be the case when the state space is infinite), and we will report results in both the finite $N$ and the infinite case. The root node of this planning tree represents the current state (or a state-action) of the MDP and its value is the maximum (over all policies defined at `MAX` nodes) of the corresponding expected sum of discounted rewards. Notice that by using a tree representation, we do not use the property that some state of the MDP can be reached by different paths (sequences of states-actions). Therefore, this state will be represented by different nodes in the tree. We could potentially merge such duplicates to form a graph instead. However, for simplicity, we choose not to merge these duplicates and keep a tree, which could make the planning problem harder. To sum up, our goal is to return, with probability $1 - \delta$, an $\varepsilon$-accurate value of the root node of this planning tree while using as low number of calls to the oracle as possible. Our contribution is an algorithm called `TrailBlazer` whose sampling strategy depends on the specific structure of the MDP and for which we provide *sample complexity* bounds in terms of a new *problem-dependent measure of the quantity of near-optimal nodes*. Before describing our contribution in more detail we first relate our setting to what has been around.

## 1.1   Related work

In this section we focus on the dependency between $\varepsilon$ and the sample complexity and all bound of the style $1/\varepsilon^c$ are up to a poly-logarithmic multiplicative factor not indicated for clarity. Kocsis and Szepesvári [12] introduced the UCT algorithm (upper-confidence bounds for trees). UCT is efficient in computer Go [7, 8, 15] and a number of other control and game problems [4]. UCT is based on generating trajectories by selecting in each `MAX` node the action that has the highest upper-confidence bound (computed according to the UCB algorithm of Auer et al. [1]). UCT converges asymptotically to the optimal solution, but its sample complexity can be worst than doubly-exponential in $(1/\varepsilon)$ for some MDPs [13]. One reason for this is that the algorithm can expand very deeply the apparently best branches but may lack sufficient exploration, especially when a narrow optimal path is hidden in a suboptimal branch. As a result, this approach works well in some problems with a specific structure but may be much worse than a uniform sampling in other problems.

On the other hand, a uniform planning approach is safe for all problems. Kearns et al. [11] generate a sparse look-ahead tree based on expanding all `MAX` nodes and sampling a finite number of children from `AVG` nodes up to a fixed depth that depends on the desired accuracy $\varepsilon$. Their sample complexity is[2] of the order of $(1/\varepsilon)^{\log(1/\varepsilon)}$, which is *non-polynomial* in $1/\varepsilon$. This bound is better than that for UCT in a worst-case sense. However, as their look-ahead tree is built in a *uniform* and *non-adaptive* way, this algorithm fails to benefit from a potentially favorable structure of the MDP.

An improved version of this sparse-sampling algorithm by Walsh et al. [17] cuts suboptimal branches in an adaptive way but unfortunately does not come with an improved bound and stays non-polynomial even in the simple Monte Carlo setting for which $K = 1$.

Although the sample complexity is certainly non-polynomial in the worst case, it *can be polynomial* in some specific problems. First, for the case of finite $N$, the sample complexity is polynomial and Szörényi et al. [16] show that a uniform sampling algorithm has complexity at most $(1/\varepsilon)^{2+\log(KN)/(\log(1/\gamma))}$. Notice that the product $KN$ represents the *branching factor* of the look-ahead planning tree. This bound could be improved for problems with specific reward structure or transition smoothness. In order to do this, we need to design non-uniform, *adaptive* algorithm that captures the possible structure of the MDP when available, while making sure that in the worst case, we do not perform worse than a uniform sampling algorithm.

The case of deterministic dynamics ($N = 1$) and rewards considered by Hren and Munos [10] has a complexity of order $(1/\varepsilon)^{(\log \kappa)/(\log(1/\gamma))}$, where $\kappa \in [1, K]$ is the branching factor of the subset of near-optimal nodes.[3] The case of stochastic rewards has been considered by Bubeck and Munos [5] but with the difference that the goal was not to approximate the optimal value function but the value of the best *open-loop* policy which consists in a sequence of actions independent of states. Their sample complexity is $(1/\varepsilon)^{\max(2,(\log \kappa)/(\log 1/\gamma))}$.

In the case of general MDPs, Buşoniu and Munos [6] consider the case of a fully known model of the MDP. For any state-action, the model returns the expected reward and the set of all next states (assuming $N$ is finite) with their corresponding transition probabilities. In that case, the complexity is $(1/\varepsilon)^{\log \kappa/(\log(1/\gamma))}$, where $\kappa \in [0, KN]$ can again be interpreted as a branching factor of the subset of near-optimal nodes. These approaches use the *optimism in the face of uncertainty* principle whose applications to planning have been have been studied by Munos [13]. `TrailBlazer` is different. It is *not optimistic* by design: To avoid voracious demand for samples it does not balance the upper-confidence bounds of all possible actions. This is crucial for polynomial sample complexity in the infinite case. The whole Section 3 shines many rays of intuitive light on this single and powerful idea.

The work that is most related to ours is `StOP` by Szörényi et al. [16] which considers the planning problem in MDPs with a generative model. Their complexity bound is of the order of $(1/\varepsilon)^{2+\log \kappa/(\log(1/\gamma))+o(1)}$, where $\kappa \in [0, KN]$ is a problem-dependent quantity. However, their $\kappa$ defined as $\lim_{\varepsilon \to 0} \max(\kappa_1, \kappa_2)$ (in their Theorem 2) is somehow difficult to interpret as a measure of the quantity of *near-optimal nodes*. Moreover, `StOP` is not computationally efficient as it requires to identify the *optimistic policy* which requires computing an upper bound on the value of *any* possible policy, whose number is exponential in the number of `MAX` nodes, which itself is exponential in the planning horizon. Although they suggest (in their Appendix F) a computational improvement, this version is not analyzed. Finally, unlike in the present paper, `StOP` does not consider the case $N = \infty$ of an unbounded number of states.

## 1.2 Our contributions

Our main result is `TrailBlazer`, an algorithm with a bound on the number of samples required to return a high-probability $\varepsilon$-approximation of the root node whether the number of next states $N$ is finite or infinite. The bounds use a problem-dependent quantity ($\kappa$ or $d$) that measures the quantity of near-optimal nodes. We now summarize the results.

__Finite__ number of next states ($N < \infty$): The sample complexity of `TrailBlazer` is of the order of[4] $(1/\varepsilon)^{\max(2,\log(N\kappa)/\log(1/\gamma)+o(1))}$, where $\kappa \in [1, K]$ is related to the branching factor of the set of near-optimal nodes (precisely defined later).

__Infinite__ number of next states ($N = \infty$): The complexity of `TrailBlazer` is $(1/\varepsilon)^{2+d}$, where $d$ is a measure of the difficulty to identify the near-optimal nodes. Notice that $d$ can be finite even if the planning problem is very challenging.[5] We also state our contributions in specific settings in comparison to previous work.

- For the case $N < \infty$, we improve over the best-known previous worst-case bound with an exponent (to $1/\varepsilon$) of $\max(2, \log(NK)/\log(1/\gamma))$ instead of $2 + \log(NK)/\log(1/\gamma)$ reported by Szörényi et al. [16].

- For the case $N = \infty$, we identify properties of the MDP (when $d = 0$) under which the sample complexity is of order (in $1/\varepsilon^2$). This is the case when there are non-vanishing action-gaps[6] from any state along near-optimal policies or when the probability of transitionning to nodes with gap $\Delta$ is upper bounded by $\Delta^2$. This complexity bound is as good as Monte-Carlo sampling and for this reason __`TrailBlazer` is a natural extension of Monte-Carlo sampling__ (where all nodes are `AVG`) __to stochastic control problems__ (where `MAX` and `AVG` nodes alternate). Also, no previous algorithm reported a polynomial bound when $N = \infty$.

- In MDPs with deterministic transitions ($N = 1$) but stochastic rewards our bound is $(1/\varepsilon)^{\max(2,\log \kappa/(\log 1/\gamma))}$ which is similar to the bound achieved by Bubeck and Munos [5] in a similar setting (open-loop policies).

- In the evaluation case without control ($K = 1$) `TrailBlazer` behaves exactly as Monte-Carlo sampling (thus achieves a complexity of $1/\varepsilon^2$), even in the case $N = \infty$.

- Finally `TrailBlazer` is easy to implement and is numerically efficient.

## 2 Monte-Carlo planning with a generative model

**Setup** We operate on a planning tree $\mathcal{T}$. Each node of $\mathcal{T}$ from the root down is alternatively either an average (`AVG`) or a maximum (`MAX`) node. For any node $s$, $\mathcal{C}[s]$ is the set of its children. We consider trees $\mathcal{T}$ for which the cardinality of $\mathcal{C}[s]$ for any `MAX` node $s$ is bounded by $K$. The cardinality $N$ of $\mathcal{C}[s]$ for any `AVG` node $s$ can be either finite, $N < \infty$, or infinite. We consider *both* cases. `TrailBlazer` applies to both situations. We provide performance guarantees for a general case and possibly tighter,

$N$-dependent guarantees in the case of $N < \infty$. We assume that we have a generative model of the transitions and rewards: Each `AVG` node $s$ is associated with a transition, a random variable $\tau_s \in \mathcal{C}[s]$ and a reward, a random variable $r_s \in [0, 1]$.

**Objective** For any node $s$, we define the value function $\mathcal{V}[s]$ as the optimum over policies $\pi$ (giving a successor to all `MAX` nodes) of the sum of discounted expected rewards playing policy $\pi$,

$$\mathcal{V}[s] = \sup_\pi \mathbb{E}\left[\sum_{t \geq 0} \gamma^t r_{s_t} \Big| s_0 = s, \pi\right],$$

where $\gamma \in (0, 1)$ is the *discount factor*. If $s$ is an `AVG` node, $\mathcal{V}$ satisfies the following Bellman equation,

$$\mathcal{V}[s] = \mathbb{E}[r_s] + \gamma \sum_{s' \in \mathcal{C}[s]} p(s'|s)\mathcal{V}[s'].$$

If $s$ is a `MAX` node, then $\mathcal{V}[s] = \max_{s' \in \mathcal{C}[s]} \mathcal{V}[s']$.

The planner has access to the oracle which can be called for any `AVG` node $s$ to either get a reward $r$ or a transition $\tau$ which are two independent random variables identically distributed as $r_s$ and $\tau_s$ respectively.

With the notation above, **our goal is to estimate the value $\mathcal{V}[s_0]$ of the root node** $s_0$ using the **smallest possible number of oracle calls**. More precisely, given any $\delta$ and $\varepsilon$, we want to output a value $\mu_{\varepsilon,\delta}$ such that $\mathbb{P}[|\mu_{\varepsilon,\delta} - \mathcal{V}[s_0]| > \varepsilon] \leq \delta$ using the smallest

possible number of oracle calls $n_{\varepsilon,\delta}$. The number of calls is the *sample complexity* of the algorithm.

---

1: **Input:** $\delta, \varepsilon$
2: **Set:** $\eta \leftarrow \gamma^{1/\max(2,\log(1/\varepsilon))}$
3: **Set:** $\lambda \leftarrow 2\log(\varepsilon(1-\gamma))^2 \frac{\log\left(\frac{\log(K)}{(1-\eta)}\right)}{\log(\gamma/\eta)}$
4: **Set:** $m \leftarrow (\log(1/\delta) + \lambda)/((1-\gamma)^2\varepsilon^2)$
5: **Use:** $\delta$ and $\eta$ as global parameters
6: **Output:**
    $\mu \leftarrow$ call the root with parameters $(m, \varepsilon/2)$

Figure 1: `TrailBlazer`

---

1: **Input:** $m, \varepsilon$
2: **Initialization:** {Only executed on first call}
3: SampledNodes $\leftarrow \emptyset$,
4: $r \leftarrow 0$
5: **Run:**
6: **if** $\varepsilon \geq 1/(1-\gamma)$ **then**
7:     **Output:** 0
8: **end if**
9: **if** $|$SampledNodes$| > m$ **then**
10:     ActiveNodes $\leftarrow$ SampledNodes$(1 : m)$
11: **else**
12:     **while** $|$SampledNodes$| < m$ **do**
13:         $\tau \leftarrow$ {new sample of next state}
14:         SampledNodes.append$(\tau)$
15:         $r \leftarrow r+$[new sample of reward]
16:     **end while**
17:     ActiveNodes $\leftarrow$ SampledNodes
18: **end if** {At this point, $|$ActiveNodes$| = m$}
19: **for** all unique nodes $s \in$ ActiveNodes **do**
20:     $k \leftarrow$ #occurrences of $s$ in ActiveNodes
21:     $\nu \leftarrow$ call $s$ with parameters $(k, \varepsilon/\gamma)$
22:     $\mu \leftarrow \mu + \nu k/m$
23: **end for**
24: **Output:** $\gamma\mu + r/|$SampledNodes$|$

Figure 2: `AVG` node

---

### 2.1 Blazing the trails with `TrailBlazer`

To fulfill the above objective, our `TrailBlazer` constructs a planning tree $\mathcal{T}$ which is, at any time, a finite subset of the potentially infinite tree. Only the already visited nodes are in $\mathcal{T}$ and explicitly represented in memory. Taking the object-oriented paradigm, each node of $\mathcal{T}$ is a persistent object with its own memory which can receive and perform calls respectively from and to other nodes. A node can potentially be called several times (with different parameters) during the run of `TrailBlazer` and may reuse (some of) its stored (transition and reward) samples. In particular, after node $s$ receives a call from its parent, node $s$ may perform internal computation by calling its own children in order to return a real value to its parent.

Pseudocode of `TrailBlazer` is in Figure 1 along with the subroutines for `MAX` nodes in Figure 3 and `AVG` nodes in Figure 2. A node (`MAX` or `AVG`) is called with two parameters $m$ and $\varepsilon$, which represent some requested properties of the returned value: $m$ controls the desired *variance* and $\varepsilon$ the desired *maximum bias*. We now describe the `MAX` and `AVG` node subroutines.

**MAX nodes** A MAX node $s$ keeps a lower and an upper bound of its children values which with high probability simultaneously hold at all times. It sequentially calls its children with different parameters in order to get more and more precise estimates of their values. Whenever the upper bound of one child becomes lower than the maximum lower bound, this child is discarded. This process can stop in two ways: 1) The set $\mathcal{L}$ of the remaining children shrunk enough such that there is a single child $b^\star$ left. In this case, $s$ calls $b^\star$ with the same parameters that $s$ received and uses the output of $b^\star$ as its own output. 2) The precision we have on the value of the remaining children is high enough. In this case, $s$ returns the highest estimate of the children in $\mathcal{L}$. Note that the MAX node is eliminating actions to identify the best. Any other best-arm identification algorithm for bandits can be adapted instead.

1: **Input:** $m, \varepsilon$
2: $\mathcal{L} \leftarrow$ all children of the node
3: $\ell \leftarrow 1$
4: **while** $|\mathcal{L}| > 1$ and $\mathtt{U} \geq (1 - \eta)\varepsilon$ **do**
5: $\quad \mathtt{U} \leftarrow \frac{2}{1-\gamma}\sqrt{\frac{\log(K\ell/(\delta\varepsilon)) + \gamma/(\eta-\gamma) + \lambda + 1}{\ell}}$
6: $\quad$ **for** $b \in \mathcal{L}$ **do**
7: $\quad\quad \mu_b \leftarrow$ call $b$ with $(\ell, \mathtt{U}\eta/(1-\eta))$
8: $\quad$ **end for**
9: $\quad \mathcal{L} \leftarrow \left\{ b : \mu_b + \frac{2\mathtt{U}}{1-\eta} \geq \sup_j \left[ \mu_j - \frac{2\mathtt{U}}{1-\eta} \right] \right\}$
10: $\quad \ell \leftarrow \ell + 1$
11: **end while**
12: **if** $|\mathcal{L}| > 1$ **then**
13: $\quad$ **Output:** $\mu \leftarrow \max_{b \in \mathcal{L}} \mu_b$
14: **else** $\{ \mathcal{L} = \{b^\star\} \}$
15: $\quad b^\star \leftarrow \arg\max_{b \in \mathcal{L}} \mu_b$
16: $\quad \mu \leftarrow$ call $b^\star$ with $(m, \eta\varepsilon)$
17: $\quad$ **Output:** $\mu$
18: **end if**

Figure 3: MAX node

**AVG nodes** Every AVG node $s$ keeps a list of all the children that it already sampled and a reward estimate $r \in \mathbb{R}$. Note that the list may contain the same child multiple times (this is particularly true for $N < \infty$). After receiving a call with parameters $(m, \varepsilon)$, $s$ checks if $\varepsilon \geq 1/(1-\gamma)$. If this condition is verified, then it returns zero. If not, $s$ considers the first $m$ sampled children and potentially samples more children from the generative model if needed. For every child $s'$ in this list, $s$ calls it with parameters $(k, \varepsilon/\gamma)$, where $k$ is the number of times a transition toward this child was sampled. It returns $r + \gamma\mu$, where $\mu$ is the average of all the children estimates.

**Anytime algorithm** TrailBlazer is *naturally anytime*. It can be called with slowly decreasing $\varepsilon$, such that $m$ is always increased only by 1, without having to throw away **any** previously collected samples. Executing TrailBlazer with $\varepsilon'$ and then with $\varepsilon < \varepsilon'$ leads to the same amount of computation as immediately running TrailBlazer with $\varepsilon$.

**Practical considerations** The parameter $\lambda$ exists so the behavior depends only on the randomness of oracle calls and the parameters $(m, \varepsilon)$ that the node has been called with. This is a desirable property because it opens the possibility to extend the algorithm to more general settings, for instance if we have also MIN nodes. However, for practical purposes, we may set $\lambda = 0$ and modify the definition of U in Figure 3 by replacing $K$ with the number of oracle calls made so far globally.

# 3 Cogs whirring behind

Before diving into the analysis we explain the ideas behind TrailBlazer and the choices made.

**Tree-based algorithm** The number of policies the planner can consider is *exponential* in the number of states. This leads to two major challenges. First, reducing the problem to multi-arm bandits on the set of the policies would hurt. When a reward is collected from a state, all the policies which could reach that state are affected. Therefore, it is useful to share the information between the policies. The second challenge is computational as it is infeasible to keep all policies in memory.

These two problems immediately vanish with just how TrailBlazer is formulated. Contrary to Szörényi et al. [16], we do not represent the policies explicitly or update them simultaneously to share the information, but we store all the information directly in the planning tree we construct. Indeed, by having all the nodes being separate entities that store their own information, we can share information between policies without explicitly having to enforce it.

We steel ourselves for the detailed understanding with the following two arguments. They shed light from two different angles on the very same key point: *Do not refine more paths than you need to!*

**Delicate treatment of uncertainty**    First, we give intuition about the *two parameters* which measure the requested *precision* of a call. The output estimate $\mu$ of any call with parameters $(m, \varepsilon)$ verifies the following property (conditioned on a high-probability event),

$$\forall \lambda \quad \mathbb{E}\left[e^{\lambda(\mu - \mathcal{V}[s])}\right] \leq \exp\left(\alpha + \varepsilon|\lambda| + \frac{\sigma^2\lambda^2}{2}\right), \text{ with } \sigma^2 = O\left(1/m\right) \text{ and constant } \alpha. \quad (1)$$

This awfully looks like the definition of $\mu$ being uncentered sub-Gaussian, except that instead of $\lambda$ in the exponential function, there is $|\lambda|$ and there is a $\lambda$-independent constant $\alpha$. Inequality 1 implies that the absolute value of the bias of the output estimate $\mu$ is bounded by $\varepsilon$,

$$\left|\mathbb{E}\left[\mu\right] - \mathcal{V}\left[s\right]\right| \leq \varepsilon.$$

As in the sub-Gaussian case, the second term $\frac{1}{2}\sigma^2\lambda^2$ is a variance term. Therefore, $\varepsilon$ controls the maximum bias of $\mu$ and $1/m$ control its sub-variance. In some cases, getting high-variance or low-variance estimate matters less as it is going to be averaged later with other independent estimates by an ancestor `AVG` node. In this case we prefer to query for high variance rather than a low one, in order to decrease sample complexity.

From $\sigma$ and $\varepsilon$ it is possible to deduce a confidence bounds on $|\mu - \mathcal{V}[s]|$ by typically summing the bias $\varepsilon$ and a term proportional to the standard deviation $\sigma = O\left(1/\sqrt{m}\right)$. Previous approaches [16, 5] consider a *single* parameter, representing the width of this high-probability confidence interval. `TrailBlazer` is different. In `TrailBlazer`, the nodes can perform high-variance and low-bias queries but can also query for both low-variance and low-bias. `TrailBlazer` treats these two types of queries *differently*. This is the whetstone of `TrailBlazer` and the reason why it is not optimistic.

**Refining few paths**    In this part we explain the condition $|\text{SampledNodes}| > m$ in Figure 2, which is crucial for our approach and results. First notice, that as long as `TrailBlazer` encounters only `AVG` nodes, it behaves just like Monte-Carlo sampling — without the `MAX` nodes we would be just doing a simple averaging of trajectories. However, when `TrailBlazer` encounters a `MAX` node it *locally* uses more samples around this `MAX` node, *temporally* moving away from a Monte-Carlo behavior. This enables `TrailBlazer` to compute the best action at this `MAX` node. Nevertheless, once this best action is identified with high probability, the algorithm should behave again like Monte-Carlo sampling. Therefore, `TrailBlazer` *forgets* the additional nodes, sampled just because of the `MAX` node, and only keeps in memory the first $m$ ones. This is done with the following line in Figure 2,

$$\text{ActiveNodes} \leftarrow \text{SampledNodes}(1 : m).$$

Again, while additional transitions were useful for some `MAX` node parents to decide which action to pick, they are discarded once this choice is made. Note that they can become useful again if an ancestor becomes unsure about which action to pick and needs more precision to make a choice. This is an important difference between `TrailBlazer` and some previous approaches like `UCT` where all the already sampled transitions are equally refined. This treatment enables us to provide *polynomial* bounds on the sample complexity for some special cases even in the infinite case ($N = \infty$).

## 4    `TrailBlazer` is good and cheap — consistency and sample complexity

In this section, we start by our consistency result, stating that `TrailBlazer` outputs a correct value in a PAC (probably approximately correct) sense. Later, we define a measure of the problem difficulty which we use to state our sample-complexity results. We remark that the following consistency result holds whether the state space is finite or infinite.

**Theorem 1.** *For all $\varepsilon$ and $\delta$, the output $\mu_{\varepsilon,\delta}$ of `TrailBlazer` called on the root $s_0$ with $(\varepsilon, \delta)$ verifies*

$$\mathbb{P}\left[|\mu_{\varepsilon,\delta} - \mathcal{V}\left[s_0\right]| > \varepsilon\right] < \delta.$$

### 4.1    Definition of the problem difficulty

We now define a measure of problem difficulty that we use to provide our sample complexity guarantees. We define a set of *near-optimal* nodes such that exploring only this set is enough to compute an optimal policy. Let $s'$ be a `MAX` node of tree $\mathcal{T}$. For any of its descendants $s$, let $c_{\rightarrow s}(s') \in \mathcal{C}\left[s'\right]$ be the child of $s'$ in the path between $s'$ and $s$. For any `MAX` node $s$, we define

$$\Delta_{\rightarrow s}(s') = \max_{x \in \mathcal{C}[s']} \mathcal{V}\left[x\right] - \mathcal{V}\left[c_{\rightarrow s}(s')\right].$$

$\Delta_{\to s}(s')$ is the difference of the sum of discounted rewards stating from $s'$ between an agent playing optimally and one playing first the action toward $s$ and then optimally.

**Definition 1** (near-optimality). *We say that a node $s$ of depth $h$ is near-optimal, if for any even depth $h'$,*

$$\Delta_{\to s}(s_{h'}) \leq 16 \frac{\gamma^{(h-h')/2}}{\gamma(1-\gamma)}$$

*with $s_{h'}$ the ancestor of $s$ of even depth $h'$. Let $\mathcal{N}_h$ be the set of all near-optimal nodes of depth $h$.*

**Remark 1.** *Notice that the subset of near-optimal nodes contains all required information to get the value of the root. In the case $N = \infty$, when $p(s|s') = 0$ for all $s$ and $s'$, then our definition of near-optimality nodes leads to the smallest subset in a sense we precise in Appendix C. We prove that with probability $1 - \delta$, `TrailBlazer` only explores near-optimal nodes. Therefore, the size of the subset of near-optimal nodes directly reflects the sample complexity of `TrailBlazer`.*

In Appendix C, we discuss the negatives of other potential definitions of near-optimality.

### 4.2 Sample complexity in the finite case

We first state our result where the set of the `AVG` children nodes is *finite* and bounded by $N$.

**Definition 2.** *We define $\kappa \in [1, K]$ as the smallest number such that*

$$\exists C \, \forall h, \quad |\mathcal{N}_{2h}| \leq C N^h \kappa^h.$$

Notice that since the total number of nodes of depth $2h$ is bounded by $(KN)^h$, $\kappa$ is upper-bounded by $K$, the maximum number of `MAX`'s children. However $\kappa$ can be as low as 1 in cases when the set of near-optimal nodes is small.

**Theorem 2.** *There exists $C > 0$ and $K$ such that for all $\varepsilon > 0$ and $\delta > 0$, with probability $1 - \delta$, the sample-complexity of `TrailBlazer` (the number of calls to the generative model before the algorithm terminates) is*

$$n(\varepsilon, \delta) \leq C(1/\varepsilon)^{\max\left(2, \frac{\log(N\kappa)}{\log(1/\gamma)} + o(1)\right)} \left(\log(1/\delta) + \log(1/\varepsilon)\right)^{\alpha},$$

*where $\alpha = 5$ when $\log(N\kappa)/\log(1/\gamma) \geq 2$ and $\alpha = 3$ otherwise.*

This provides a problem-dependent sample-complexity bound, which already in the worst case ($\kappa = K$) improves over the best-known worst-case bound $\widetilde{\mathcal{O}}\left((1/\varepsilon)^{2 + \log(KN)/\log(1/\gamma)}\right)$ [16]. This bound gets better as $\kappa$ gets smaller and is minimal when $\kappa = 1$. This is, for example, the case when the gap (see definition given in Equation 2) at `MAX` nodes is uniformly lower-bounded by some $\Delta > 0$. In this case, this theorem provides a bound of order $(1/\varepsilon)^{\max(2, \log(N)/\log(1/\gamma))}$. However, we will show in Remark 2 that we can further improve this bound to $(1/\varepsilon)^2$.

### 4.3 Sample complexity in the infinite case

Since the previous bound depends on $N$, it does not apply to the infinite case with $N = \infty$. We now provide a sample complexity result in the case $N = \infty$. However, notice that when $N$ is bounded, then *both* results apply.

We first define gap $\Delta(s)$ for any `MAX` node $s$ as the difference between the best and second best arm,

$$\Delta(s) = \mathcal{V}[i^{\star}] - \max_{i \in \mathcal{C}[s], i \neq i^{\star}} \mathcal{V}[i] \quad \text{with } i^{\star} = \arg\max_{i \in \mathcal{C}[s]} \mathcal{V}[i]. \tag{2}$$

For any even integer $h$, we define a random variable $S^h$ taking values among `MAX` nodes of depth $h$, in the following way. First, from every `AVG` nodes from the root to nodes of depth $h$, we draw a single transition to one of its children according to the corresponding transition probabilities. This defines a subtree with $K^{h/2}$ nodes of depth $h$ and we choose $S^h$ to be one of them *uniformly at random*. Furthermore, for any even integer $h' < h$ we note $S^h_{h'}$ the `MAX` node ancestor of $S^h$ of depth $h'$.

**Definition 3.** *We define $d \geq 0$ as the smallest $d$ such that for all $\xi$ there exists $a > 0$ for which for all even $h > 0$,*

$$\mathbb{E}\left[K^{h/2}\mathbb{1}\left(S^h \in \mathcal{N}_h\right)\prod_{\substack{0 \leq h' < h \\ h' \equiv 0 (\mathrm{mod}\ 2)}}\left(\frac{\xi}{\gamma^{h-h'}}\right)^{\mathbb{1}\left(\Delta(S_{h'}^h) < 16\frac{\gamma^{(h-h')/2}}{\gamma(1-\gamma)}\right)}\right] \leq a\gamma^{-dh}$$

*If no such $d$ exists, we set $d = \infty$.*

This definition of $d$ takes into account the size of the near-optimality set (just like $\kappa$) but unlike $\kappa$ it also takes into account the difficulty to identify the near-optimal paths.

Intuitively, the expected number of oracle calls performed by a given `AVG` node $s$ is proportional to: $(1/\varepsilon^2) \times$ (the product of the inverted squared gaps of the set of `MAX` nodes in the path from the root to $s$) $\times$ (the probability of reaching $s$ by following a policy which always tries to reach $s$).

Therefore, a near-optimal path with a larger number of small `MAX` node gaps can be considered *difficult*. By assigning a larger weight to difficult nodes, we are able to give a better characterization of the actual complexity of the problem and provide polynomial guarantees on the sample complexity for $N = \infty$ when $d$ is finite.

**Theorem 3.** *If $d$ is finite then there exists $C > 0$ such that for all $\varepsilon > 0$ and $\delta > 0$, the expected sample complexity of* `TrailBlazer` *satisfies*

$$\mathbb{E}\left[n(\varepsilon, \delta)\right] \leq C\frac{(\log(1/\delta) + \log(1/\varepsilon))^3}{\varepsilon^{2+d}}.$$

Note that this result holds in expectation only, contrary to Theorem 2 which holds in high probability.

We now give an example for which $d = 0$, followed by a special case of it.

**Lemma 1.** *If there exists $c > 0$ and $b > 2$ such that for any near-optimal* `AVG` *node $s$,*

$$\mathbb{P}\left[\Delta\left(\tau_s\right) \leq x\right] \leq cx^b,$$

*where the random variable $\tau_s$ is a successor state from $s$ drawn from the MDP's transition probabilities, then $d = 0$ and consequently the sample complexity is of order $1/\varepsilon^2$.*

**Remark 2.** *If there exists $\Delta_{\min}$ such that for any near-optimal* `MAX` *node $s$, $\Delta(s) \geq \Delta_{\min}$ then $d = 0$ and the sample complexity is of order $1/\varepsilon^2$. Indeed, in this case as $\mathbb{P}\left[\Delta_s \leq x\right] \leq (x/\Delta_{\min})^b$ for any $b > 2$ for which $d = 0$ by Lemma 1.*

## 5 Conclusion

We provide a new Monte-Carlo planning algorithm `TrailBlazer` that works for MDPs where the number of next states $N$ can be either finite or infinite. `TrailBlazer` is easy to implement and is numerically efficient. It comes packaged with a PAC consistency and two problem-dependent sample-complexity guarantees expressed in terms of a measure (defined by $\kappa$) of the quantity of near-optimal nodes or a measure (defined by $d$) of the difficulty to identify the near-optimal paths. The sample complexity of `TrailBlazer` improves over previous worst-case guarantees. What's more, `TrailBlazer` exploits MDPs with specific structure by exploring only a fraction of the whole search space when either $\kappa$ or $d$ is small. In particular, we showed that if the set of near-optimal nodes have non-vanishing action-gaps, then the sample complexity is $\widetilde{\mathcal{O}}(1/\varepsilon^2)$, which is the same rate as Monte-Carlo sampling. This is a pretty decent evidence that `TrailBlazer` is a natural extension of Monte-Carlo sampling to stochastic control problems.

**Acknowledgements**   The research presented in this paper was supported by French Ministry of Higher Education and Research, Nord-Pas-de-Calais Regional Council, a doctoral grant of École Normale Supérieure in Paris, Inria and Carnegie Mellon University associated-team project EduBand, and French National Research Agency projects ExTra-Learn (n.ANR-14-CE24-0010-01) and BoB (n.ANR-16-CE23-0003)

## Footnotes

*on the leave from SequeL team, INRIA Lille - Nord Europe, France

[2]neglecting exponential dependence in $\gamma$

[3]nodes that need to be considered in order to return a near-optimal approximation of the value at the root

[4] neglecting logarithmic terms in $\varepsilon$ and $\delta$

[5] since when $N = \infty$ the actual branching factor of the set of reachable nodes is infinite

[6] defined as the difference in values of best and second-best actions

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
