[Supplementary Material]

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

```
1: Input: δ, ε
2: Set: η ← γ^(1/ max(2,log(1/ε)))
3: Set: λ ← 2 log(ε(1 − γ))² log(log(K)/(1−η)) / log(γ/η)
4: Set: m ← (log(1/δ) + λ)/((1 − γ)²ε²)
5: Use: δ and η as global parameters
6: Output:
       μ ← call the root with parameters (m, ε/2)
```

Figure 1: TrailBlazer

**Objective** For any node $s$, we define the value function $\mathcal{V}[s]$ as the optimum over policies $\pi$ (giving a successor to all MAX nodes) of the sum of discounted expected rewards playing policy $\pi$,

$$\mathcal{V}[s] = \sup_{\pi} \mathbb{E}\left[ \sum_{t \geq 0} \gamma^t r_{s_t} \Big| s_0 = s, \pi \right],$$

where $\gamma \in (0, 1)$ is the *discount factor*. If $s$ is an AVG node, $\mathcal{V}$ satisfies the following Bellman equation,

$$\mathcal{V}[s] = \mathbb{E}[r_s] + \gamma \sum_{s' \in \mathcal{C}[s]} p(s'|s)\mathcal{V}[s'].$$

If $s$ is a MAX node, then $\mathcal{V}[s] = \max_{s' \in \mathcal{C}[s]} \mathcal{V}[s']$.

The planner has access to the oracle which can be called for any AVG node $s$ to either get a reward $r$ or a transition $\tau$ which are two independent random variables identically distributed as $r_s$ and $\tau_s$ respectively.

With the notation above, **our goal is to estimate the value $\mathcal{V}[s_0]$ of the root node $s_0$** using the **smallest possible number of oracle calls**. More precisely, given any $\delta$ and $\varepsilon$, we want to output a value $\mu_{\varepsilon,\delta}$ such that $\mathbb{P}[|\mu_{\varepsilon,\delta} - \mathcal{V}[s_0]| > \varepsilon] \leq \delta$ using the smallest possible number of oracle calls $n_{\varepsilon,\delta}$. The number of calls is the *sample complexity* of the algorithm.

```
1: Input: m, ε
2: Initialization: {Only executed on first call}
3: SampledNodes ← ∅,
4: r ← 0
5: Run:
6: if ε ≥ 1/(1 − γ) then
7:     Output: 0
8: end if
9: if |SampledNodes| > m then
10:     ActiveNodes ← SampledNodes(1 : m)
11: else
12:     while |SampledNodes| < m do
13:         τ ← {new sample of next state}
14:         SampledNodes.append(τ)
15:         r ← r+[new sample of reward]
16:     end while
17:     ActiveNodes ← SampledNodes
18: end if {At this point, |ActiveNodes| = m}
19: for all unique nodes s ∈ ActiveNodes do
20:     k ← #occurrences of s in ActiveNodes
21:     ν ← call s with parameters (k, ε/γ)
22:     μ ← μ + νk/m
23: end for
24: Output: γμ + r/|SampledNodes|
```

Figure 2: AVG node

### 2.1 Blazing the trails with TrailBlazer

To fulfill the above objective, our TrailBlazer constructs a planning tree $\mathcal{T}$ which is, at any time, a finite subset of the potentially infinite tree. Only the already visited nodes are in $\mathcal{T}$ and explicitly represented in memory. Taking the object-oriented paradigm, each node of $\mathcal{T}$ is a persistent object with its own memory which can receive and perform calls respectively from and to other nodes. A node can potentially be called several times (with different parameters) during the run of TrailBlazer and may reuse (some of) its stored (transition and reward) samples. In particular, after node $s$ receives a call from its parent, node $s$ may perform internal computation by calling its own children in order to return a real value to its parent.

Pseudocode of TrailBlazer is in Figure 1 along with the subroutines for MAX nodes in Figure 3 and AVG nodes in Figure 2. A node (MAX or AVG) is called with two parameters $m$ and $\varepsilon$, which represent some requested properties of the returned value: $m$ controls the desired *variance* and $\varepsilon$ the desired *maximum bias*. We now describe the MAX and AVG node subroutines.

**MAX nodes** A MAX node $s$ keeps a lower and an upper bound of its children values which with high probability simultaneously hold at all times. It sequentially calls its children with different parameters in order to get more and more precise estimates of their values. Whenever the upper bound of one child becomes lower than the maximum lower bound, this child is discarded. This process can stop in two ways: 1) The set $\mathcal{L}$ of the remaining children shrunk enough such that there is a single child $b^\star$ left. In this case, $s$ calls $b^\star$ with the same parameters that $s$ received and uses the output of $b^\star$ as its own output. 2) The precision we have on the value of the remaining children is high enough. In this case, $s$ returns the highest estimate of the children in $\mathcal{L}$. Note that the MAX node is eliminating actions to identify the best. Any other best-arm identification algorithm for bandits can be adapted instead.

```
1: Input: m, ε
2: L ← all children of the node
3: ℓ ← 1
4: while |L| > 1 and U ≥ (1 − η)ε do
5:     U ← (2/(1−γ)) √((log(Kℓ/(δε))+γ/(η−γ)+λ+1)/ℓ)
6:     for b ∈ L do
7:         μ_b ← call b with (ℓ, Uη/(1 − η))
8:     end for
9:     L ← { b : μ_b + 2U/(1−η) ≥ sup_j [μ_j − 2U/(1−η)] }
10:     ℓ ← ℓ + 1
11: end while
12: if |L| > 1 then
13:     Output: μ ← max_{b∈L} μ_b
14: else { L = {b^⋆} }
15:     b^⋆ ← arg max_{b∈L} μ_b
16:     μ ← call b^⋆ with (m, ηε)
17:     Output: μ

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

# A  Consistency

First, we comment on the behavior of the nodes when called multiple times.

**Remark 3.** *Notice that the output of the call to a node $s$ with parameters $(m, \varepsilon)$ and global $\delta$ does not depend on when this call was performed. That means if this call is performed again later in the execution with the same parameters $(m, \varepsilon)$ and $\delta$, the output would be the same. This is because, first, the* MAX *nodes does not store anything so that their behavior only depend on $s, m, \varepsilon$, and $\delta$. Second, the* AVG *nodes store the already sampled rewards, transitions and the order they were sampled but when called with parameters $(m, \varepsilon)$ and $\delta$, it returns the average of the $m$ first sampled children only.*

From now we consider any $\delta, \eta$, and $\gamma$, such that $0 < \gamma < \eta < 1$ and $0 < \delta < 1/2$.

We define a random variable $\mu_s(m, \varepsilon)$ to be the output of the node $s$ called with parameters $(m, \varepsilon)$ and global parameters $\delta$ and $\eta$. Notice that following Remark 3, $\mu_s(m, \varepsilon)$ does depend on $s$, $m$, $\varepsilon$, $\delta$, $\eta$ and all the transitions and rewards of itself and its descendants but does not depend on the current state of the algorithm.

We define the function $\mathtt{U}(m, \varepsilon)$ such that the value $\mathtt{U}(m, \varepsilon) + \varepsilon$ is the width of a high-probability confidence interval for $\mu_s(m, \varepsilon)$ which is used in MAX node,

$$\mathtt{U}(m, \varepsilon) \stackrel{\text{def}}{=} \frac{2}{1 - \gamma} \sqrt{\frac{\log(K\ell/(\delta\varepsilon)) + \gamma/(\eta - \gamma) + 1}{m}}.$$

We define $\ell_{\max}(\varepsilon) \stackrel{\text{def}}{=} \max\{\ell \ : \ \mathtt{U}(\ell, \varepsilon) \geq (1 - \eta)\varepsilon\}$. By definition of TrailBlazer, this is the maximum value that the children of a MAX node would be ever called with. We also define $\overline{\ell}(m, \varepsilon) \stackrel{\text{def}}{=} \max(m, \ell_{\max}(\varepsilon))$

We now construct event $\mathcal{B}_s$ for which Lemma 2 proves that it holds with high probability and that on $\mathcal{B}_s$, TrailBlazer works well.

**Definition 4.** *For any node $s$, $\varepsilon > 0$, and $m \in \mathbb{N}^+$,*

- *If $\varepsilon \geq 1/(1 - \gamma)$,*

$$\mathcal{B}_s(m, \varepsilon) \stackrel{\text{def}}{=} \Omega.$$

- *Else, if $s$ is a* MAX *node,*

$$\mathcal{B}_s(m, \varepsilon) \stackrel{\text{def}}{=} \cap_{i \in \mathcal{C}[s]} \left( \cap_{\ell \leq \ell_{\max}(\varepsilon)} C_{i,\ell}(\varepsilon) \cap \mathcal{B}_{i,\ell_{\max}} \right)(\varepsilon)$$

*with* $\begin{cases} C_{i,\ell}(\varepsilon) \stackrel{\text{def}}{=} \left\{ |\mu_i\left(\ell, \frac{\eta}{1-\eta}\mathtt{U}(\ell,\varepsilon)\right) - \mathcal{V}[i]| \leq \left(1 + \frac{\eta}{1-\eta}\right)\mathtt{U}(\ell,\varepsilon) \right\} \\ \mathcal{B}_{i,\ell}(\varepsilon) \stackrel{\text{def}}{=} \mathcal{B}_i\left(\ell, \frac{\eta}{1-\eta}\mathtt{U}(\ell,\varepsilon)\right). \end{cases}$

- *Else, if $s$ is an* AVG *node,*

$$\mathcal{B}_s(m, \varepsilon) \stackrel{\text{def}}{=} \cap_{k \in \{1,..,m\}} \mathcal{B}_{\tau_{s,k}}(\nu_s(k,m), \varepsilon/\gamma),$$

*with $\nu_s(k,m) \stackrel{\text{def}}{=} |\{j : \tau_{s,k} = \tau_{s,j} \text{ and } j \leq m\}|$.*

Intuitively, $C$ is the event where the confidence bounds are satisfied. Using events $C$, Definition 4 recursively constructs the events $\mathcal{B}$ needed for Lemma 2.

We remark that there is no loop in this recursive definition as the definition of $\mathcal{B}_s$ only uses $\{\mathcal{B}_i\}_i$ where $i$ is a child of $s$. Moreover, this way we define $\mathcal{B}_s$ in a finite number of steps as going down through MAX and AVG nodes increases $\varepsilon$ to $\varepsilon\eta/\gamma$ and reaches $1/(1 - \gamma)$ in a finite number of steps.

Moreover, note that $\mathcal{B}_s$ is *not* an intersection of a finite number of $C_{i,\ell}$ because $\tau_{s,k}$ is a random variable and therefore in the AVG case, $\mathcal{B}_s$ is not just the intersection of $m$ events.

**Remark 4.** *Notice that, by induction, for any $\delta > 0, \eta > 0, \varepsilon_1, \varepsilon_2, m_1, m_2$, and any node $s$,*

$$\varepsilon_1 \leq \varepsilon_2 \text{ and } m_1 \geq m_2 \implies \mathcal{B}_s(m_1, \varepsilon_1) \subseteq \mathcal{B}_s(m_2, \varepsilon_2).$$

**Lemma 2.** *For any $k \in \mathbb{N}^+$ and any node $s$, integers $m' \geq m$, and $\varepsilon > 0$, such that $\delta < \left(\overline{\ell}(m, \eta\varepsilon)\right)^{-h_s(\varepsilon)}$ and*

- *either $k$ is odd, $s$ is an `AVG` node, and $\varepsilon > \frac{(\gamma/\eta)^{(k-1)/2}}{1-\gamma}$,*

- *or $k$ is even, $s$ is a `MAX` node, and $\varepsilon > \frac{(\gamma/\eta)^{(k/2)-1}}{\eta(1-\gamma)}$,*

*then the two following properties are verified*

**Property A:** $\quad \mathbb{P}\left[\mathcal{B}_s(m, \varepsilon)\right] \geq 1 - \delta \dfrac{\left(\overline{\ell}(m, \varepsilon) K/\eta^2\right)^{h_s(\varepsilon)} - 1}{K - 1}$

**Property B:** $\quad \forall \lambda, \mathbb{E}\left[e^{\lambda(\mu_s(m, \varepsilon) - \mathcal{V}[s])} \big| \mathcal{B}_s(m', \varepsilon)\right] \leq \exp\left(2\delta \left(\dfrac{\overline{\ell}(m, \varepsilon)}{\eta^2}\right)^{h_s(\varepsilon)} \times \dfrac{1 - \varepsilon\theta(s)}{\eta/\gamma - 1} + \varepsilon|\lambda| + \dfrac{\lambda^2}{2(1-\gamma)^2 m}\right),$

*where $\theta(s) = \begin{cases} 1 - \gamma & \text{if } s \text{ is a `MAX` node} \\ (1-\gamma)/\gamma & \text{if } s \text{ is an `AVG` node} \end{cases}$ and $\quad h_s(\varepsilon) = \max\left(0, \frac{\log(\theta(s)\gamma\varepsilon)}{\log(\gamma/\eta)}\right)$*

Lemma 2 claims that on a high-probability event $\mathcal{B}$, the sub-variance of $\mu_s$ is of order at most $1/m$ and its bias of order at most $\varepsilon$ up to the term independent of $\lambda$. We prove it by induction on $k$.

*Proof.* $1°$ Let $k = 1$. Then, $k$ is odd and $s$ is an `AVG` node. Since $\varepsilon > 1/(1 - \gamma)$, $s$ outputs zero by the definition. For any $m' \geq m$ we have $\mathcal{B}_s(m, \varepsilon) = \mathcal{B}_s(m', \varepsilon) = \Omega$, since $\varepsilon > 1/(1 - \gamma)$.

Property A: By definition $h_s(\varepsilon) \geq 0$, therefore $\delta\frac{\left(\overline{\ell}(m,\varepsilon)K/\eta^2\right)^{h_s(\varepsilon)}-1}{K-1} \geq 0$. Then,

$$\mathbb{P}\left[\Omega\right] = 1 \geq 1 - \delta\frac{\left(\overline{\ell}(m, \varepsilon) K/\eta^2\right)^{h_s(\varepsilon)} - 1}{K - 1}.$$

Property B: First, as the rewards are bounded from above by one, we have $\mathcal{V}[s] \leq \sum_h \gamma^h \leq 1/(1-\gamma)$. Second, $\mathcal{V}[s] \geq 0$ as the rewards are bounded from below by zero. Finally, $\varepsilon\theta(s) \leq 1$, hence

$$2\delta\left(\frac{\overline{\ell}(m, \varepsilon)}{\eta^2}\right)^{h_s(\varepsilon)} \times \frac{1 - \varepsilon\theta(s)}{\eta/\gamma - 1} \geq 0.$$

We combine these bounds to get

$$\forall \lambda, \mathbb{E}\left[e^{\lambda(\mu_s(m, \varepsilon) - \mathcal{V}[s])}\Big|\mathcal{B}_s(m, \varepsilon)\right] = \mathbb{E}\left[e^{\lambda(\mu_s(m, \varepsilon) - \mathcal{V}[s])}\right] = \mathbb{E}\left[e^{\lambda(0 - \mathcal{V}[s])}\right] \leq \exp\left(\frac{|\lambda|}{1 - \gamma}\right)$$

$$\leq \exp\left(\varepsilon|\lambda|\right) \leq \exp\left(2\delta\left(\frac{\overline{\ell}(m, \varepsilon)}{\eta^2}\right)^{h_s(\varepsilon)} \times \frac{1 - \varepsilon\theta(s)}{\eta/\gamma - 1} + \varepsilon|\lambda| + \frac{\lambda^2}{2(1-\gamma)^2 m}\right).$$

$2°$ Let $k \in \mathbb{N}^+$. Assume that Lemma 2 is true for any $k' \leq k - 1$ and let us prove it for $k$. We prove it separately for $k$ odd and even.

$k$ **is even:** Let $s$ be a `MAX` node, $m$ any integer and $\varepsilon$ such that

$$\varepsilon > \frac{(\gamma/\eta)^{(k/2)-1}}{\eta(1-\gamma)} \quad \text{and} \quad \frac{1}{\overline{\ell}(m, \varepsilon)} > \delta.$$

We deduce that

$$\eta\varepsilon > \eta\frac{(\gamma/\eta)^{(k/2)-1}}{\eta(1-\gamma)} = \frac{(\gamma/\eta)^{(k/2)-1}}{1-\gamma}.$$

Also, any child of $s$ is an `AVG` node as nodes alternate between `AVG` and `MAX`. We can thus apply the induction assumption to any child of $s$ with parameters $(m', \varepsilon')$ with $\varepsilon' < \eta\varepsilon$ and $m' \leq \overline{\ell}(m, \varepsilon)$.

For any child $i \in \mathcal{C}[s]$ of $s$ and any integer $\ell > 0$, we remind the reader the definition of $\mathcal{B}_{i,\ell}$,

$$\mathcal{B}_{i,\ell} = \mathcal{B}_i\left(\ell, \frac{\eta}{1-\eta}\mathtt{U}(\ell,\varepsilon)\right).$$

If $\mathtt{U}(\ell,\varepsilon) \geq (1-\eta)\varepsilon$, then $\frac{\eta}{1-\eta}\mathtt{U}(\ell,\varepsilon) \geq \eta\varepsilon$ and $\ell \leq \ell_{\max}(\varepsilon)$.

We can thus apply our induction assumption to get that for any $\ell \leq \ell_{\max}$,

$$\text{A } \mathbb{P}[\mathcal{B}_{i,\ell}] \geq 1 - \delta\frac{\left(\bar{\ell}\left(\ell, \frac{\eta}{1-\eta}\mathtt{U}(\ell,\varepsilon)\right)K/\eta^2\right)^{h_i\left(\frac{\eta}{1-\eta}\mathtt{U}(\ell,\varepsilon)\right)} - 1}{K - 1} \tag{3}$$

$$\text{B } \forall\lambda, \mathbb{E}\left[e^{\lambda\left(\mu_i(\ell,\frac{\eta}{1-\eta}\mathtt{U}(\ell,\varepsilon))-\mathcal{V}[i]\right)}\middle|\mathcal{B}_{i,\ell}\right] \tag{4}$$

$$\leq \exp\left(2\delta\left(\frac{\bar{\ell}\left(\ell, \frac{\eta}{1-\eta}\mathtt{U}(\ell,\varepsilon)\right)}{\eta^2}\right)^{h_i\left(\frac{\eta}{1-\eta}\mathtt{U}(\ell,\varepsilon)\right)} \times \frac{1 - \frac{\eta}{1-\eta}\mathtt{U}(\ell,\varepsilon)\theta(i)}{\eta/\gamma - 1} + \frac{\eta}{1-\eta}\mathtt{U}(\ell,\varepsilon)|\lambda| + \frac{\lambda^2}{2(1-\gamma)^2\ell}\right)$$

We first prove Property A.

Combining Inequality 4 with Lemma 6, we get for all $i \in \mathcal{C}[s]$ and $\ell \leq \ell_{\max}$,

$$\mathbb{P}[C_{i,\ell}{}^\mathsf{c}|\mathcal{B}_{i,\ell}] \leq 2\exp\left(2\delta\ell_{\max}(\eta\varepsilon)^{h_i(\eta\varepsilon)} \times \frac{1 - \varepsilon\theta(i)}{\eta/\gamma - 1}\right)\exp\left(-\frac{\left(\mathtt{U}(\ell,\varepsilon)\left(1 + \frac{\eta}{1-\eta}\right) - \frac{\eta}{1-\eta}\mathtt{U}(\ell,\varepsilon)\right)^2}{\frac{2}{(1-\gamma)^2\ell}}\right)$$

$$\leq 2\exp\left(2\frac{1 - \varepsilon\theta(i)}{\eta/\gamma - 1}\right)\exp\left(-\frac{(1-\gamma)^2\ell}{2}\mathtt{U}(\ell,\varepsilon)^2\right)$$

$$\text{as } (\delta < 1/\ell) \leq 2\exp\left(\frac{2\gamma}{\eta - \gamma}\right)\exp\left(-\frac{(1-\gamma)^2\ell}{2}\frac{1}{(1-\gamma)^2}\frac{4\log\left(K\ell/(\delta\varepsilon(1-\gamma))\right) + 4\gamma/(\eta-\gamma) + 4}{\ell}\right)$$

$$\leq \frac{1}{2}\left(\frac{\delta\varepsilon(1-\gamma)}{K\ell}\right)^2 \leq \frac{1}{2}\frac{\delta\varepsilon(1-\gamma)}{K\ell^2}.$$

Using a union bound over all $\ell \leq \ell_{\max}$ we get

$$\mathbb{P}[\cup_{\ell \leq \ell_{\max}}C_{i,\ell}{}^\mathsf{c}|\mathcal{B}_{i,\ell}] \leq \sum_l \frac{1}{2}\frac{\delta\varepsilon(1-\gamma)}{K\ell^2} \leq \frac{\pi^2}{12}\frac{\delta\varepsilon(1-\gamma)}{K} \leq \frac{\delta\varepsilon(1-\gamma)}{K}.$$

Now, we can finally bound the event for node $s$ as

$$\begin{aligned}
\mathbb{P}[\mathcal{B}_s(m,\varepsilon)] &= \mathbb{P}\left[\cap_{i\in\mathcal{C}[s]}\left(\cap_{\ell\leq\ell_{\max}}C_{i,\ell} \cap \mathcal{B}_{i,\ell_{\max}}\right)\right] \\
&= 1 - \mathbb{P}\left[\cup_{i\in\mathcal{C}[s]}\left(\cup_{\ell\leq\ell_{\max}}C_{i,\ell}{}^\mathsf{c} \cup \mathcal{B}_{i,\ell_{\max}}{}^\mathsf{c}\right)\right] \\
&\geq 1 - \sum_{i\in\mathcal{C}[s]}\mathbb{P}\left[\cup_{\ell\leq\ell_{\max}}C_{i,\ell}{}^\mathsf{c} \cup \mathcal{B}_{i,\ell_{\max}}{}^\mathsf{c}\right] \\
&= 1 - \sum_{i\in\mathcal{C}[s]}\left(1 - \mathbb{P}[\cap_{\ell\leq\ell_{\max}}C_{i,\ell} \cap \mathcal{B}_{i,\ell_{\max}}]\right) \\
&= 1 - \sum_{i\in\mathcal{C}[s]}\left(1 - \mathbb{P}[\cap_{\ell\leq\ell_{\max}}C_{i,\ell}|\mathcal{B}_{i,\ell_{\max}}]\mathbb{P}[\mathcal{B}_{i,\ell_{\max}}]\right).
\end{aligned}$$

We use Inequality 3 to get

$$\mathbb{P}[\mathcal{B}_{i,\ell_{\max}}] \geq 1 - \delta\frac{\left(\bar{\ell}(\ell_{\max},\eta\varepsilon)K/\eta^2\right)^{h_i(\eta\varepsilon)} - 1}{K - 1}.$$

To relate $h_s(\varepsilon)$ to its child, we realize that $i$ is an `AVG` node and hence,

$$
\begin{aligned}
h_i(\eta\varepsilon) &= \max\left(0, \frac{\log\left(\theta(s)\gamma\eta\varepsilon(1-\gamma)\right)}{\log\left(\gamma/\eta\right)}\right) \\
&= \frac{\log\left(\theta(s)\gamma\eta\varepsilon(1-\gamma)\right)}{\log\left(\gamma/\eta\right)} \\
&= \frac{\log\left(\eta\varepsilon(1-\gamma)\right)}{\log\left(\gamma/\eta\right)} \\
&= \frac{\log(\eta/\gamma) + \log\left((1/\gamma)\varepsilon(1-\gamma)\right)}{\log\left(\gamma/\eta\right)} \\
&= h_s(\varepsilon) - 1.
\end{aligned}
$$

We go back to bounding $\mathbb{P}\left[\mathcal{B}_s(m,\varepsilon)\right]$,

$$
\begin{aligned}
\mathbb{P}\left[\mathcal{B}_s(m,\varepsilon)\right] &\geq 1 - \sum_{i\in\mathcal{C}[s]}\left(1 - \left(\frac{\delta\varepsilon(1-\gamma)}{K}\right)\left(1 - \delta\frac{\left(\overline{\ell}\left(\ell_{\max},\eta\varepsilon\right)K/\eta^2\right)^{h_i(\eta\varepsilon)}-1}{K-1}\right)\right) \\
&\geq 1 - \sum_{i\in\mathcal{C}[s]}\left(\frac{\delta}{K} + \delta\frac{\left(\overline{\ell}\left(\ell_{\max},\varepsilon\right)K\right)^{h_s(\varepsilon)-1}-1}{K-1}\right) \qquad (\text{as } \varepsilon \leq 1/(1-\gamma)) \\
&= 1 - \delta\left(1 + K\frac{\left(\overline{\ell}\left(\ell_{\max},\varepsilon\right)K\right)^{h_s(\varepsilon)-1}-1}{K-1}\right) \\
&\geq 1 - \delta\frac{\left(\overline{\ell}\left(m,\varepsilon\right)K/\eta^2\right)^{h_s(\varepsilon)}-1}{K-1}.
\end{aligned}
$$

To prove Property B, we introduce additional notation. We define

$$
\mu_{i,\ell} \stackrel{\text{def}}{=} \mu_i\left(\ell, \tfrac{\eta}{1-\eta}\mathtt{U}(\ell,\varepsilon)\right).
$$

We also define the empirically best and the best child as

$$
\widehat{\imath}_\ell \stackrel{\text{def}}{=} \arg\max_{i\in\mathcal{C}[s]}\mu_{i,\ell} \quad\text{and}\quad i^\star \stackrel{\text{def}}{=} \arg\max_{i\in\mathcal{C}[s]}\mathcal{V}[i].
$$

We define the random variable $\mathcal{L}_\ell$ as the set of all children that would be kept in Line 9 in Figure 3 if they are called with parameter $\ell$ as

$$
\mathcal{L}_\ell \stackrel{\text{def}}{=} \left\{i\in\mathcal{C}[s] : \mu_{i,\ell} + \frac{2\mathtt{U}(\ell,\varepsilon)}{1-\eta} \geq \sup_j\left[\mu_{j,\ell} - \frac{2\mathtt{U}(\ell,\varepsilon)}{1-\eta}\right]\right\}.
$$

Notice that the `MAX` node is not constructing sets $\{\mathcal{L}_\ell\}_\ell$ iteratively: The reason is that some of the children might have been eliminated in the earlier rounds. However, at the end of the while loop in Figure 3, we get set $\mathcal{L}$ that is the intersection of all $\{\mathcal{L}_\ell\}_\ell$,

$$
\mathcal{L} = \cap_{\ell\leq\ell_{\max}}\mathcal{L}_\ell.
$$

We define $\mathcal{G}_\ell$ (which is *not* a random variable) as

$$
\mathcal{G}_\ell \stackrel{\text{def}}{=} \left\{i\in\mathcal{C}[s] : \mathcal{V}[i] + \frac{\mathtt{U}(\ell,\varepsilon)}{1-\eta} \geq \mathcal{V}[i^\star] - \frac{\mathtt{U}(\ell,\varepsilon)}{1-\eta}\right\}. \tag{5}
$$

For simplicity, we also define $\widehat{\imath} \stackrel{\text{def}}{=} \widehat{\imath}_{\ell_{\max}}$, $\mathcal{G} \stackrel{\text{def}}{=} \mathcal{G}_{\ell_{\max}}$, and for all $i\in\mathcal{C}[s]$, $\mu_i \stackrel{\text{def}}{=} \mu_{i,\ell_{\max}}$.

We can now prove Property B. By definition of $C_{i,\ell}$ and induction assumption, we have

$$
\left\{\cap_{i\in\mathcal{C}[s],\ell\leq\ell_{\max}}C_{i,\ell}\right\} \subseteq \left\{\forall i,\ell : |\mu_{i,\ell} - \mathcal{V}[i]| \leq \mathtt{U}(\ell,\varepsilon)/(1-\eta)\right\}
$$

and we set $C \stackrel{\text{def}}{=} \cap_{i\in\mathcal{C}[s],\ell\leq\ell_{\max}}C_{i,\ell}$.

We distinguish two cases depending on the cardinality of the *deterministic* set $\mathcal{G}$ and assume $C$.

**First case: $|\mathcal{G}| = 1$.** If $|\mathcal{L}| > 1$ then for all $i \in \mathcal{L}$ such that $i \notin \mathcal{G} = \{i^\star\}$

$$\mu_i \leq \mathcal{V}[i] + \varepsilon \qquad\qquad \text{(because } C_{i,\ell_{\max}} \text{ holds)}$$
$$< V(i^\star) - \varepsilon \qquad\qquad \text{(because } i \notin \mathcal{G}_s = \{i^\star\})$$
$$\leq \mu_{i^\star}^{s,t} \qquad\qquad \text{(because } C_{i^\star,\ell_{\max}} \text{ holds)}$$

Therefore $\arg\max_{i \in \mathcal{L}} \mu_i = i^\star$ which implies that $i^\star$ is called with parameters $(m', \varepsilon')$ with

$$\varepsilon' \leq \eta\varepsilon \leq \varepsilon \quad \text{and} \quad m' \geq m.$$

If $|\mathcal{L}| = 1$, then $\mathcal{L} = \{i^\star\}$ and consequently $i^\star$ is called with parameters $(m, \varepsilon)$.

In both cases, on event $C$, $\mu_s(m, \varepsilon)$ is the output of a call to $i^\star$ with parameters $(m', \varepsilon')$ with $m' \geq m$ and $\varepsilon' \leq \varepsilon$ and $m' \leq \ell_{\max}$. We use Property B of induction assumption as $\ell_{\max} \geq m$ and $\mathtt{U}(\ell_{\max}, \varepsilon)\eta/(1 - \eta) \leq \varepsilon$.

$$\forall \lambda, \mathbb{E}\left[e^{\lambda\left(\mu_{i^\star}(m',\varepsilon') - \mathcal{V}[i^\star]\right)} \big| B_{i^\star,\ell_{\max}}\right] = \mathbb{E}\left[e^{\lambda(\mu_s(m,\varepsilon) - \mathcal{V}[s])} \big| B_{i^\star,\ell_{\max}}\right]$$

$$\leq \exp\left(2\delta\left(\frac{\overline{\ell}(m',\varepsilon')}{\eta^2}\right)^{h_{i^\star}(\varepsilon')} \times \frac{1 - \varepsilon'\theta(i^\star)}{\eta/\gamma - 1} + \varepsilon'|\lambda| + \frac{\lambda^2}{2(1-\gamma)^2 m'}\right)$$

$$\leq \exp\left(2\delta\left(\frac{\overline{\ell}(m,\eta\varepsilon)}{\eta^2}\right)^{h_s(\eta\varepsilon)} \times \frac{1 - \eta\varepsilon\theta(s)}{\eta/\gamma - 1} + \varepsilon|\lambda| + \frac{\lambda^2}{2(1-\gamma)^2 m}\right)$$

Also, because for all $i, j \in \mathcal{C}[s]$ with $i \neq j$, $B_i$ and $\mu_j$ are independent, we can claim that

$$\forall \lambda, \mathbb{E}\left[e^{\lambda(\mu_s(m,\varepsilon) - \mathcal{V}[s])} \big| \cap_{i \in \mathcal{C}[s]} \mathcal{B}_{i,\ell_{\max}}\right] \leq \exp\left(2\delta\left(\frac{\overline{\ell}(m,\eta\varepsilon)}{\eta^2}\right)^{h_s(\eta\varepsilon)} \times \frac{1 - \eta\varepsilon\theta(s)}{\eta/\gamma - 1} + \varepsilon|\lambda| + \frac{\lambda^2}{2(1-\gamma)^2 m}\right).$$

Using Lemma 7 we finaly get for all $\lambda$

$$\mathbb{E}\left[e^{\lambda(\mu_s(m,\varepsilon) - \mathcal{V}[s])} \big| \mathcal{B}_s(m,\varepsilon)\right] = \mathbb{E}\left[e^{\lambda(\mu_s(m,\varepsilon) - \mathcal{V}[s])} \big| \cap_{i \in \mathcal{C}[s]} \left(\cap_{\ell \leq \ell_{\max}} C_{i,\ell} \cap \mathcal{B}_{i,\ell_{\max}}\right)\right]$$

$$\leq \frac{\mathbb{E}\left[e^{\lambda(\mu_s(m,\varepsilon) - \mathcal{V}[s])} \big| \cap_{i \in \mathcal{C}[s]} \mathcal{B}_{i,\ell_{\max}}\right]}{\mathbb{P}\left[\cap_{i \in \mathcal{C}[s], \ell \leq \ell_{\max}} C_{i,\ell} \big| \cap_{i \in \mathcal{C}[s]} \mathcal{B}_{i,\ell_{\max}}\right]}$$

$$\leq \frac{\exp\left(2\delta\left(\frac{\overline{\ell}(m,\eta\varepsilon)}{\eta^2}\right)^{h_s(\eta\varepsilon)} \times \frac{1 - \eta\varepsilon\theta(s)}{\eta/\gamma - 1} + \varepsilon|\lambda| + \frac{\lambda^2}{2(1-\gamma)^2 m}\right)}{1 - \delta\varepsilon(1 - \gamma)}$$

To simplify the expression, we note that for $0 < x \leq 1/2$, we have that $1/(1-x) \leq 1 + 2x \leq \exp(2x)$. We also use that $\delta < 1/2$.

$$\frac{\exp\left(2\delta\left(\frac{\overline{\ell}(m,\eta\varepsilon)}{\eta^2}\right)^{h_s(\eta\varepsilon)} \times \frac{1 - \eta\varepsilon\theta(s)}{\eta/\gamma - 1}\right)}{1 - \delta\varepsilon(1 - \gamma)} \leq \exp\left(2\delta\left(\frac{\overline{\ell}(m,\eta\varepsilon)}{\eta^2}\right)^{h_s(\eta\varepsilon)} \times \frac{1 - \eta\varepsilon\theta(s)}{\eta/\gamma - 1} + 2\delta\varepsilon(1 - \gamma)\right)$$

$$\leq \exp\left(2\delta\left(\frac{\overline{\ell}(m,\varepsilon)}{\eta^2}\right)^{h_s(\varepsilon)} \left(\frac{1 - \eta\varepsilon(1-\gamma)/\gamma}{\eta/\gamma - 1} + \varepsilon(1 - \gamma)\right)\right)$$

$$= \exp\left(2\delta\left(\frac{\overline{\ell}(m,\varepsilon)}{\eta^2}\right)^{h_s(\varepsilon)} \frac{1 - \varepsilon(1 - \gamma)}{\eta/\gamma - 1}\right)$$

$$= \exp\left(2\delta\left(\frac{\overline{\ell}(m,\varepsilon)}{\eta^2}\right)^{h_s(\varepsilon)} \frac{1 - \varepsilon\theta(s)}{\eta/\gamma - 1}\right)$$

**Second case:** $|\mathcal{G}| > 1$.    For all $i \in \mathcal{G}_l$,

$$
\begin{aligned}
\mu_{i,\ell} &\geq \mathcal{V}[i] - \mathtt{U}(\ell,\varepsilon)/(1-\eta) && \text{(because $C$ holds)}\\
&\geq V(i^\star) - 3\mathtt{U}(\ell,\varepsilon)/(1-\eta) && \text{(because $i \in \mathcal{G}_l$)}\\
&\geq \mu_{i^\star}^{s,t} - 4\mathtt{U}(\ell,\varepsilon)/(1-\eta) && \text{(because $C$ holds)}
\end{aligned}
$$

Therefore for all $\ell \leq \ell_{\max}$,

$$\mathcal{G}_\ell \subset \mathcal{L}_\ell. \tag{6}$$

As a result $|\mathcal{L}| > |\mathcal{G}| > 1$. The output is then the maximum of the estimates in $\mathcal{L}$. The best estimate $\mu_{\widehat{\imath}}$ in $\mathcal{L}$ verifies

$$
\begin{aligned}
\mu_{\widehat{\imath}} &\geq \mu_{i^\star} && \text{(by definition of $\widehat{\imath}$)}\\
&\geq V(i^\star) - \mathtt{U}(\ell_{\max},\varepsilon)/(1-\eta) && \text{(because $C$ holds)}.
\end{aligned}
$$

It also verifies that

$$
\begin{aligned}
\mu_{\widehat{\imath}} &\leq V(\widehat{\imath}) + \mathtt{U}(\ell_{\max},\varepsilon)/(1-\eta) && \text{(because $C_{\widehat{\imath},\ell_{\max}}$ holds)}\\
&\leq V(i^\star) + \mathtt{U}(\ell_{\max},\varepsilon)/(1-\eta) && \text{(by definition of $i^\star$)}.
\end{aligned}
$$

Since $\mu_s(m,\varepsilon) = \mu_{\widehat{\imath}}$, we have

$$\big|\mu_s(m,\varepsilon) - \mathcal{V}[s]\big| \leq \frac{\mathtt{U}(\ell_{\max},\varepsilon)}{1-\eta} \leq \varepsilon.$$

It follows that

$$
\mathbb{E}\left[e^{\lambda(\mu_s(m,\varepsilon)-\mathcal{V}[s])}\big|\mathcal{B}_s(m,\varepsilon)\right] \leq \exp\left(|\lambda|\varepsilon\right)
$$
$$
\leq \exp\left(2\delta\left(\frac{\bar{\ell}(m,\varepsilon,)}{\eta^2}\right)^{h_s(\varepsilon,)}\times\frac{1-\varepsilon,\theta(s)}{\eta/\gamma-1} + |\lambda|\varepsilon + \frac{\lambda^2}{2(1-\gamma)^2m}\right).
$$

$k$ **is odd:**    Let $s$ be a $\mathtt{AVG}$ node, $m \in \mathbb{N}_0^+$, and $\varepsilon$ such that

$$\varepsilon > \frac{(\gamma/\eta)^{(k-1)/2}}{(1-\gamma)} \quad\text{and}\quad \frac{1}{\max(m,\ell_{\max}(\varepsilon))} > \delta.$$

Hence, we can deduce

$$\frac{\varepsilon}{\gamma} > \frac{(\gamma/\eta)^{(k-1)/2}}{\gamma(1-\gamma)} = \frac{(\gamma/\eta)^{\frac{k-1}{2}-1}}{\eta(1-\gamma)}.$$

Also, any child of $s$ is a $\mathtt{MAX}$ node as nodes alternate between $\mathtt{AVG}$ and $\mathtt{MAX}$. We can thus apply induction assumption to any child of $s$ with parameters $(m',\varepsilon')$ with $\varepsilon' \geq \varepsilon/\gamma$ and $m' \leq \max(m,\ell_{\max}(\varepsilon))$.

For any child $i \in \mathcal{C}[s]$ of $s$ and any integer $\ell \leq m$ we can thus apply our induction assumption to get

$$
\mathbb{P}[\mathcal{B}_i(l,\varepsilon/\gamma)] \geq 1 - \delta\frac{\left(\bar{\ell}(l,\varepsilon/\gamma)\,K/\eta^2\right)^{h_i(\varepsilon/\gamma)}-1}{K-1}
$$

$$
\forall\lambda, \mathbb{E}\left[e^{\lambda(\mu_i(\ell,\varepsilon/\gamma))-\mathcal{V}[i])}\bigg|\mathcal{B}_i(\ell,\varepsilon/\gamma)\right] \leq \exp\left(2\delta\left(\frac{\bar{\ell}(\ell,\varepsilon/\gamma)}{\eta^2}\right)^{h_i(\varepsilon/\gamma)}\times\frac{1-\varepsilon/\gamma\theta(i)}{\eta/\gamma-1} + |\lambda|\varepsilon/\gamma + \frac{\lambda^2}{2(1-\gamma)^2\ell}\right)
$$

We state again the definition of $\mathcal{B}_s$

$$\mathcal{B}_s(m,\varepsilon) \overset{\text{def}}{=} \cap_{i\in\mathcal{C}[s]}\mathcal{B}_i(\nu_i(m),\varepsilon/\gamma)$$

with $\nu_i(m) \overset{\text{def}}{=} |\{j : \tau_{s,j} = i \text{ and } j \leq m\}|$.

We first prove property A.

$$\mathbb{P}\left[\cup_{i\in\mathcal{C}[s]}\mathcal{B}_i(\nu_i(m),\varepsilon/\gamma)^\mathsf{c}\right] \leq \sum_{i\in\mathcal{C}[s]}\mathbb{P}\left[\mathcal{B}_i(\nu_i(m),\varepsilon/\gamma)^\mathsf{c}\right]$$

$$= \int_k \sum_{i\in\mathcal{C}[s]}\mathbb{P}\left[\mathcal{B}_i(\nu_i(m),\varepsilon/\gamma)^\mathsf{c}|\nu_i(m)=k\right]d\pi(k)$$

$$= \int_k \sum_{i\in\mathcal{C}[s]}\mathbb{P}\left[\mathcal{B}_i(k_i,\varepsilon/\gamma)^\mathsf{c}|\nu_i(m)=k_i\right]d\pi(k)$$

$$= \int_k \sum_{i\in\mathcal{C}[s]}\mathbb{P}\left[\mathcal{B}_i(k_i,\varepsilon/\gamma)^\mathsf{c}\right]d\pi(k)$$

$$\leq \int_k \sum_{i\in\mathcal{C}[s];k_i>0}\delta\frac{\left(\overline{\ell}\left(k_i,\varepsilon/\gamma\right)K/\eta^2\right)^{h_i(\varepsilon/\gamma)}-1}{K-1}\pi(k)$$

$$\leq \int\delta\frac{\left(\overline{\ell}\left(k_i,\varepsilon\right)K/\eta^2\right)^{h(\varepsilon)}-1}{K-1}\pi(k)$$

$$= \delta\frac{\left(\overline{\ell}\left(k_i,\varepsilon\right)K/\eta^2\right)^{h(\varepsilon)}-1}{K-1}$$

We now prove property B.

$$\mathbb{E}\left[\exp\left((\lambda/m)\sum_{i\in\mathcal{C}[s]}\nu_i(m)\mu_i(\nu_i(m),\varepsilon/\gamma)\right)\Big|\mathcal{B}_s(m,\varepsilon)\right]$$

$$= \int_k\mathbb{E}\left[\exp\left((\lambda/m)\sum_{i\in\mathcal{C}[s]}\nu_i(m)\mu_i(\nu_i(m),\varepsilon/\gamma)\right)\Big|\mathcal{B}_s(m,\varepsilon),(\nu_i(m))=(k_i)\right]d\pi(k)$$

$$= \int_k\mathbb{E}\left[\exp\left((\lambda/m)\sum_{i\in\mathcal{C}[s]}k_i\mu_i(k_i,\varepsilon/\gamma)\right)\Big|\mathcal{B}_s(m,\varepsilon)\right]d\pi(k)$$

$$= \int_k\mathbb{E}\left[\prod_{i\in\mathcal{C}[s]}\exp\left((\lambda/m)k_i\mu_i(k_i,\varepsilon/\gamma)\right)\Big|\mathcal{B}_s(m,\varepsilon)\right]d\pi(k)$$

$$\leq \int_k\exp\left(\sum_{i\in\mathcal{C}[s]}k_i\left(2\delta\left(\frac{\overline{\ell}\left(k_i,\varepsilon/\gamma\right)}{\eta^2}\right)^{h_i(\varepsilon/\gamma)}\times\frac{1-\varepsilon/\gamma\theta(i)}{\eta/\gamma-1}+|(\lambda/m)|\varepsilon/\gamma+\frac{(\lambda/m)^2}{2(1-\gamma)^2k_i}\right)\right)d\pi(k)$$

$$= \int_k\exp\left(m2\delta\left(\frac{\overline{\ell}\left(m,\varepsilon/\gamma\right)}{\eta^2}\right)^{h_i(\varepsilon/\gamma)}\times\frac{1-\varepsilon/\gamma\theta(i)}{\eta/\gamma-1}+|\lambda|\varepsilon/\gamma+\frac{\lambda^2}{2(1-\gamma)^2m}\right)d\pi(k)$$

$$\leq \exp\left(2\delta\left(\frac{\overline{\ell}\left(m,\varepsilon\right)}{\eta^2}\right)^{h_s(\varepsilon)}\times\frac{1-\varepsilon\theta(s)}{\eta/\gamma-1}+|\lambda|\varepsilon/\gamma+\frac{\lambda^2}{2(1-\gamma)^2m}\right)$$

We can finally bound

$$
\mathbb{E}\left[\exp\left(\lambda r + \gamma(\lambda/m)\sum_{i\in\mathcal{C}[s]}\nu_i(m)\mu_i(\nu_i(m),\varepsilon/\gamma)\right)\Big|\mathcal{B}_s(m,\varepsilon)\right]
$$

$$
\leq \exp\left(\frac{\lambda^2}{2m} + 2\delta\left(\frac{\overline{\ell}(m,\varepsilon)}{\eta^2}\right)^{h_s(\varepsilon)}\times\frac{1-\varepsilon\theta(s)}{\eta/\gamma-1} + |\lambda|\varepsilon + \frac{\lambda^2\gamma^2}{2(1-\gamma)^2m}\right)
$$

$$
\leq \exp\left(2\delta\left(\frac{\overline{\ell}(m,\varepsilon)}{\eta^2}\right)^{h_s(\varepsilon)}\times\frac{1-\varepsilon\theta(s)}{\eta/\gamma-1} + |\lambda|\varepsilon + \left(\frac{\lambda\gamma}{\sqrt{2m}(1-\gamma)} + \frac{\lambda}{\sqrt{2m}}\right)^2\right)
$$

$$
= \exp\left(2\delta\left(\frac{\overline{\ell}(m,\varepsilon)}{\eta^2}\right)^{h_s(\varepsilon)}\times\frac{1-\varepsilon\theta(s)}{\eta/\gamma-1} + |\lambda|\varepsilon + \frac{\lambda^2}{2(1-\gamma)^2m}\right).
$$

$\square$

Now we stat and prove Theorem 4 that is equivalent to Theorem 1.

**Theorem 4.** *Let $\mu$ be the output of* `TrailBlazer`*.For all $\delta' > 0, \varepsilon > 0$, there exists an event $\mathcal{B}$ such that $\mathbb{P}[\mathcal{B}] \geq 1 - \delta'$ and on $\mathcal{B}$, $|\mu - \mathcal{V}[s_0]| \leq \varepsilon$.*

*Proof.* We apply Lemma 2 to the root node $s_0$. Let us note the following:

$$
m = (\log(1/\delta) + \lambda)/((1-\gamma)^2\varepsilon^2)
$$

$$
\lambda = 2\log(\varepsilon(1-\gamma))^2\frac{\log\left(\frac{\log(K)}{(1-\eta)}\right)}{\log(\gamma/\eta)}
$$

We set $\delta = \exp(\lambda)\delta'$. We apply Lemma 2 on $s_0$ with $(m, \varepsilon/2)$. We take

$$
\mathcal{B} = \mathcal{B}_{s_0}(\log(1/\delta) + \lambda)/((1-\gamma)^2\varepsilon^2).
$$

We check that

$$
\delta\frac{\left(\overline{\ell}(m,\varepsilon)K/\eta^2\right)^{h_s(\varepsilon)} - 1}{K - 1} \leq \delta'
$$

and apply Lemma 6 with property $B$ of Lemma 2 to get

$$
|\mu_{s_0}(m,\varepsilon/2) - \mathcal{V}[s_0]| \leq \varepsilon.
$$

$\square$

# B  Sample complexity

**Definition 5** (node classification). *A node $s$ of depth $h$ is said to be*

- *suboptimal if there exists $h' < h$, such that $\Delta_{\to s}(s_{h'}) > 16\frac{\gamma^{(h-h')/2}}{\gamma(1-\gamma)}$,*

- *optimal if for all $h' < h, \Delta_{\to s}(s_{h'}) = 0$,*

- *near-optimal in all other cases.*

**Lemma 3.** *For any node $s$ of depth $h$, integer $m$, and $\varepsilon > 0$, a call to $s$ with $(m, \varepsilon)$ does not generate call to nodes of depth $h' > h + h_{\max}(\varepsilon)$ with*

$$h_{\max}(\varepsilon) \leq \left\lceil 2\frac{\log(\varepsilon(1-\gamma))}{\log(\gamma/\eta)} \right\rceil.$$

`MAX` and `AVG` nodes alternate. First, Any `AVG` node called with $\varepsilon$ can only perform calls with $\varepsilon' \geq \varepsilon/\gamma$. Second, `MAX` nodes called with $\varepsilon$ can only perform calls with $\varepsilon' \geq \eta\varepsilon$. Finally, an `AVG` node called with $\varepsilon = 1/(1-\gamma)$ *cannot* perform a call to deeper node.

For any node $s$ called with different $(m_t, \varepsilon_t)$ during the algorithm execution, we let $\Gamma_s = \max_t m_t$ be the maximum $m_t$ that $s$ has been called with.

**Lemma 4.** *We denote the parameters the root has been called with by $(m_0, \varepsilon_0)$.*

*If $\mathcal{B}_{s_0}(m, \varepsilon)$ holds then for all nodes $s$ of depth $h$ and $h' < h$, the following holds,*

$$s \text{ is suboptimal} \implies \Gamma_s = 0$$

$$\Delta(s_{h'}) > 16\frac{\gamma^{(h-h')/2}}{\gamma(1-\gamma)} \implies \text{any call to } s \text{ was generated by a call from } s_{h'} \text{ with } m \leq \Gamma_{s_{h'}}.$$

*Proof.* For any node $s$ of depth $h$, we assume that $\Gamma_s > 0$ and we want to prove that $s$ is not suboptimal. Let $h'$ be some integer with $h' < h$. We note $(m_{s_{h'}}, \varepsilon_{s_{h'}})$ the parameters $s_{h'}$ has been called with. If $\Gamma_s > 0$, then, by Lemma 3, that means that $s_{h'}$ performs a call to its child $i_s$ in direction to $s$ with parameters $(m, \varepsilon)$ such that $h_{\max}(\varepsilon) \geq h - h'$.

For any $m', \varepsilon'$, if $s_{h'}$ performs a call to $i_s$ with parameters $(m', \varepsilon')$ with $\varepsilon' \leq \varepsilon$ then

1. either $i_s \in \mathcal{L}_{\ell_{\max}(m_{s_{h'}}, \varepsilon_{s_{h'}})}$ and $\eta\varepsilon_{s_{h'}} \leq \varepsilon$,

2. or $i_s \in \mathcal{L}_\ell$ for some $\ell$ such that $\frac{\eta}{1-\eta}\mathsf{U}_\ell \leq \varepsilon$.

We use the definition of $\mathcal{G}$ (5) and the result of Equation 6.

Case 1. For all $\ell'$ we have $\mathcal{L}_{\ell'} \subset \mathcal{G}_{\ell'}$ therefore $\mathcal{V}[i^\star] - \mathcal{V}[i_s] \leq \frac{4\mathsf{U}_\ell}{1-\eta} \leq 4\frac{\varepsilon}{\eta}$.

Case 2. By definition of $\ell_{\max}$, $\mathsf{U}_{\ell_{\max}+1} \leq (1-\eta)\varepsilon_{s_{h'}}$. We also have that

$$\forall \ell' \exists a \geq 0 \quad \frac{\mathsf{U}_{\ell'+1}}{\mathsf{U}_{\ell'}} = \sqrt{\frac{\ell'}{\ell'+1}\frac{\log(\ell'+1)+a}{\log(\ell')+a}} \geq \sqrt{\frac{\ell'}{\ell'+1}} \geq 1/\sqrt{2}.$$

We put it together using $i_s \in \mathcal{L}_{\ell_{\max}} \subset \mathcal{H}_{\ell_{\max}}$,

$$\mathcal{V}[i^\star] - \mathcal{V}[i_s] \leq \frac{4\mathsf{U}_{\ell_{\max}}}{1-\eta} \leq \mathsf{U}_{\ell_{\max}} \leq \frac{4\sqrt{2}}{1-\eta}\mathsf{U}_{\ell_{\max}+1} \leq 4\sqrt{2}\varepsilon_s \leq 4\sqrt{2}\varepsilon/\eta.$$

Either $\mathcal{V}[i^\star] - \mathcal{V}[i_s] \leq \frac{4\varepsilon}{\eta}$ or $\mathcal{V}[i^\star] - \mathcal{V}[i_s] \leq \frac{4\varepsilon\sqrt{2}}{\eta}$ from which we deduce that

$$\mathcal{V}[i^\star] - \mathcal{V}[i_s] \leq \frac{4\varepsilon\sqrt{2}}{\eta}.$$

We use $h - h' \leq h_{\max}(\varepsilon) \leq 2\frac{\log(\varepsilon(1-\gamma))}{\log(\gamma/\eta)} + 1$ to get $(\gamma/\eta)^{(h-h')/2} \geq \varepsilon(1-\gamma)\gamma/\eta$ and using $\eta \leq 1$

$$\varepsilon \leq \frac{1}{\gamma(1-\gamma)}(1/\eta)^{(h-h')/2}\gamma^{(h-h')/2}.$$

As $h - h' \leq h_{\max}(\varepsilon_0)$,

$$\log\left((1/\eta)^{(h-h')/2}\right) \leq \log\left((1/\eta)^{h_{\max}(\varepsilon_0)/2}\right) \leq \frac{\log 1/\gamma}{\log(1/\varepsilon)} \frac{\log(1/\varepsilon)}{\log(\eta/\gamma)} \leq 1.$$

We combine

$$\varepsilon \leq \frac{e}{\gamma(1-\gamma)}\gamma^{(h-h')/2} \text{ and } \mathcal{V}[i^\star] - \mathcal{V}[i_s] \leq \frac{4\varepsilon\sqrt{2}}{\eta},$$

to finally get

$$\Delta_{\to s}(s_{h'}) = \mathcal{V}[i^\star] - \mathcal{V}[i_s] \leq \frac{4e\sqrt{2}}{\gamma(1-\gamma)}\gamma^{(h-h')/2} \leq 16\frac{\gamma^{(h-h')/2}}{\gamma(1-\gamma)}.$$

This is true for any $h' < h$ which proves that $s$ is not suboptimal.

We now assume that $\Delta(s_{h'}) > 16\frac{\gamma^{(h-h')/2}}{\gamma(1-\gamma)}$ and prove that if a call to $i^\star$ later generated calls to $s$, then this call to $i^\star$ was done with $(m, \varepsilon)$ with $m \leq \Gamma_{s_{h'}}$.

By Lemma 3, $h_{\max}(\varepsilon) \geq h - h'$. We distinguish three cases,

1. either $s_{h'}$ calls $i^\star$ within the while loop,

2. or $s_{h'}$ calls $i^\star$ outside the while loop with $|\mathcal{L}| > 1$,

3. or $s_{h'}$ calls $i^\star$ outside the while loop with $|\mathcal{L}| = 1$.

Case 1. If this call is made to $i \neq i^\star$ with $\varepsilon = \frac{\eta}{1-\eta}U_\ell$, with the same reasoning as above we conclude that $\mathcal{V}[i^\star] - \mathcal{V}[i_s] \leq \frac{4e\sqrt{2}}{\gamma(1-\gamma)}\gamma^{h-h'}$ which contradicts the assumption.

Case 2. We use again the same reasoning as above with $U_{\ell_{\max}}$ to conclude that if $\varepsilon_{s_{h'}} \geq \eta\varepsilon$ then $\mathcal{V}[i^\star] - \mathcal{V}[i_s] \leq \frac{4e\sqrt{2}}{\gamma(1-\gamma)}\gamma^{h-h'}$.

Case 3. In this last case, the call is done with $(m_{s_{h'}}, \varepsilon_{s_{h'}})$ and $m_{s_{h'}} \leq \Gamma_{s_{h'}}$.

$\square$

## B.1  Finite case: Proof of Theorem 2

*Proof.* Any near-optimal node of depth $h$ called with parameters $(m', \varepsilon')$ generates at most

$$\frac{\log(1/(\varepsilon\delta))}{(1-\eta)^2(1-\gamma)^2\varepsilon'^2} \quad \text{samples.}$$

This follows from the same reasoning as in previous theorem. Then we can use that $\varepsilon' \leq \varepsilon(\eta/\gamma)^{h/2}$. Therefore, the maximum number of samples $T$ is

$$T \leq \sum_{h \leq h_{\max}} N^h \kappa^h \frac{\log(1/(\varepsilon\delta))}{(1-\eta)^2(1-\gamma)^2\varepsilon^2\eta^{h/2}/\gamma^{h/2}}.$$

If $N\kappa/\gamma^2 > 1$, then,

$$T \leq \mathcal{O}\left(h_{\max}(N\kappa\eta/\gamma)^{h_{\max}} \frac{\log(1/(\varepsilon\delta))}{(1-\eta)^2(1-\gamma)^2\varepsilon^2}\right)$$

$$= \mathcal{O}\left(h_{\max}(1/\varepsilon)^{(\log(N\kappa)+2\log(\eta/\gamma))/\log(\eta/\gamma)} \frac{\log(1/(\varepsilon\delta))}{(1-\eta)^2(1-\gamma)^2\varepsilon^2}\right)$$

$$= \mathcal{O}\left(\frac{\log(1/\varepsilon)}{\log(\eta/\gamma)}(1/\varepsilon)^{\log(N\kappa)/\log(\eta/\gamma)} \frac{\log(1/(\varepsilon\delta))}{(1-\eta)^2(1-\gamma)^2}\right)$$

$$= \mathcal{O}\left((1/\varepsilon)^{\log(N\kappa)/\log(1/\gamma)+o(1)} \frac{\log(1/(\varepsilon\delta))\log(1/\varepsilon)^4}{(1-\gamma)^5}\right) \qquad \text{Setting } \eta = \gamma^{1/\max(2,\log(1/\varepsilon))}$$

$$= \mathcal{O}\left((1/\varepsilon)^{\log(N\kappa)/\log(1/\gamma)+o(1)} \left(\frac{\log(1/\delta)+\log(1/\varepsilon)}{1-\gamma}\right)^5\right)$$

If $N\kappa/\gamma^2 < 1$, then,

$$T \leq \mathcal{O}\left((1/\varepsilon)^2 \frac{1}{1-N\kappa/\gamma^2} \frac{\log(1/(\varepsilon\delta))}{(1-\eta)^2(1-\gamma)^2}\right)$$

$$= \mathcal{O}\left((1/\varepsilon)^2 \frac{(\log(1/\delta)+\log(1/\varepsilon))^3}{(1-N\kappa/\gamma^2)(1-\gamma)^4}\right). \qquad \text{Setting } \eta = \gamma^{1/\max(2,\log(1/\varepsilon))}$$

If $N\kappa/\gamma^2 = 1$, then,

$$T \leq \mathcal{O}\left(h_{\max}(1/\eta)^{h_{\max}} \frac{\log(1/(\varepsilon\delta))}{(1-\eta)^2(1-\gamma)^2\varepsilon^2}\right)$$

$$T \leq \mathcal{O}\left(\frac{\log(1/(\varepsilon\delta))\log(1/\varepsilon)^4}{(1-\gamma)^5\varepsilon^2}\right) \qquad (1/\eta)^{h_{\max}} = \mathcal{O}(1)$$

$$T \leq \mathcal{O}\left((1/\varepsilon)^2 \left(\frac{\log(1/\delta)+\log(1/\varepsilon)}{1-\gamma}\right)^5\right).$$

$\square$

## B.2 Infinite case

For any MAX node $s'$ of depth $h'$ and any $h > h'$, we define $\Gamma^h_{s'}$ as the maximum $m$ the node $s'$ has been called with; such that this call generated calls up to depth $h$.

**Lemma 5.** *There exists $\xi$ such that for any node $s$ of depth $h$ and any ancestor $s_{h'}$ of $s$ of depth $h' \leq h$,*

$$\mathbb{E}\left[\Gamma_s\right] \leq \mathbb{1}\left(s \text{ is not suboptimal}\right) p(s|s_0) \prod_{\substack{0 \leq h' < h \\ h' \equiv 0 (\text{mod } 2)}} \left(\frac{1}{\xi\gamma^{h-h'}}\right)^{\mathbb{1}\left(\Delta(s_{h'}) < 16\frac{\gamma^{(h-h')/2}}{\gamma(1-\gamma)}\right)} \Gamma_{s_0}.$$

*Proof.* We prove that

$$\mathbb{E}\left[\Gamma^h_{s_{h'+1}}\right] \leq \mathbb{1}\left(s_{h'+1} \text{ is not suboptimal}\right) \mathbb{E}\left[\Gamma_{s^h_{h'}}\right] p(s|s') \left(\frac{1}{\xi\gamma^{h-h'}}\right)^{\mathbb{1}\left(\Delta(s_{h'}) < 16\frac{\gamma^{(h-h')/2}}{\gamma(1-\gamma)}\right)}.$$

If $\Delta_{s_{h'}} \geq 16\frac{\gamma^{(h-h')/2}}{\gamma(1-\gamma)}$ then by Lemma 4, either $s$ is suboptimal and therefore $\Gamma_s = 0$ or the calls of $s_{h'}$ towards $s$ which generate calls up to depth $h$ are done with parameters $m \leq \mathbb{E}\left[\Gamma^h_s\right]$.

In the other case, a node called with $(m, \varepsilon)$ generates at most $\mathcal{O}\left(1/\varepsilon^2\right)$ samples. Moreover, since $s_{h'}$ is of depth $h'$, we know that it is called with $\varepsilon \geq \mathcal{O}\left((\gamma/\eta)^{(h-h')/2}\right) \geq \xi\gamma^{(h-h')/2}/K$ for some $\xi$ as $(1/\eta)^{h_{\max}} = \mathcal{O}(1)$.

Also, if no sample from $s_{h'}$ reaches $s_{h'+1}$ then $\Gamma_s = 0$. Together, we get that

$$\Gamma_{s_{h'+1}} \leq \mathbb{1}\left( \text{ One of the } \left(\frac{1}{\xi\gamma^{h-h'}}\right)^{\mathbb{1}\left(\Delta(s_{h'})<16\frac{\gamma^{(h-h')/2}}{\gamma(1-\gamma)}\right)} \text{ samples from } s_{h'} \text{ goes to } s_{h'+1} \right) \Gamma_{s_{h'}}$$

$$\mathbb{E}\left[\Gamma_{s_{h'+1}}\right] \leq \mathbb{P}\left[ \text{ One of the } \left(\frac{1}{\xi\gamma^{h-h'}}\right)^{\mathbb{1}\left(\Delta(s_{h'})<16\frac{\gamma^{(h-h')/2}}{\gamma(1-\gamma)}\right)} \text{ samples from } s_{h'} \text{ goes to } s_{h'+1} \right] \mathbb{E}\left[\Gamma_{s_{h'}}\right]$$

$$\leq \mathbb{E}\left[\Gamma_{s^h_{h'}}\right] p(s|s') \left(\frac{1}{\xi\gamma^{h-h'}}\right)^{\mathbb{1}\left(\Delta(s_{h'})<16\frac{\gamma^{(h-h')/2}}{\gamma(1-\gamma)}\right)}.$$

Now $\mathbb{E}\left[\Gamma_{s_{h'+1}}\right] = 0$ and therefore $s_{h'+1}$ is $h$-suboptimal. Moreover, if one of the $s_{h'}$ is $h$-suboptimal then $s$ is suboptimal. To finish the proof, we multiply all these inequalities over $h'$. $\square$

### B.2.1 Proof of Theorem 3

*Proof.* We bound the expected number of oracle calls as

$$\mathbb{E}\left[n(\varepsilon, \delta)\right] \leq \mathbb{E}\left[\sum_{h=1}^{h_{\max}(\varepsilon)} \sum_{s \text{ of depth } h} \Gamma_s\right]$$

$$= \sum_{h=1}^{h_{\max}(\varepsilon)} \sum_{s \text{ of depth } h} \mathbb{E}\left[\Gamma_s\right]$$

$$\leq \sum_{h=1}^{h_{\max}(\varepsilon)} K^{h/2}\mathbb{E}\left[ \mathbb{1}\left(S^h \in \mathcal{N}_h\right) \prod_{\substack{0 \leq h' < h \\ h' \equiv 0 (\text{mod } 2)}} \left(\frac{1}{\xi\gamma^{2(h-h')}}\right)^{\mathbb{1}\left(\Delta(S^h_{h'})<16\frac{\gamma^{(h-h')/2}}{\gamma(1-\gamma)}\right)} \right] \Gamma_{s_0}$$

$$\leq \sum_{h=1}^{h_{\max}(\varepsilon)} a\gamma^{-dh}\Gamma_{s_0}$$

$$\leq ah_{\max}(\varepsilon) \mathcal{O}\left(1/\varepsilon\right)^d \Gamma_{s_0}$$

$$\leq \mathcal{O}\left( \log(1/\varepsilon)(1/\varepsilon)^d \frac{(\log(1/\delta) + \log(1/\varepsilon))^2}{\varepsilon^2} \right)$$

$$\leq \mathcal{O}\left( \frac{(\log(1/\delta) + \log(1/\varepsilon))^3}{\varepsilon^{2+d}} \right).$$

This is true on $\mathcal{B}_s(m, \varepsilon)$ which holds with probability $1 - \delta$ but as $\delta \ll \mathcal{O}\left(1/\varepsilon^3\right)$, the result also holds on the whole probability space. $\square$

### B.2.2 Proof of Lemma 1

*Proof.* We note $x_{h'} \stackrel{\text{def}}{=} \frac{1}{\xi\gamma^{h-h'}}$ and $y_{h'} \stackrel{\text{def}}{=} 16\frac{\gamma^{(h-h')/2}}{\gamma(1-\gamma)}$ so that

$$\mathbb{E}\left[ \mathbb{1}\left(S^h \in \mathcal{N}_h\right) \prod_{\substack{0 \leq h' < h \\ h' \equiv 0 (\text{mod } 2)}} \left(\frac{1}{\xi\gamma^{h-h'}}\right)^{\mathbb{1}\left(\Delta(S^h_{h'})<16\frac{\gamma^{(h-h')/2}}{\gamma(1-\gamma)}\right)} \right] = \mathbb{E}\left[ \mathbb{1}\left(S^h \in \mathcal{N}_h\right) \prod_{\substack{0 \leq h' < h \\ h' \equiv 0 (\text{mod } 2)}} x_{h'}^{\mathbb{1}\left(\Delta(S^h_{h'})<y_{h'}\right)} \right].$$

Since at all `AVG` nodes transitions are taken independently:

$$\mathbb{E}\left[\mathbb{1}\left(S^h \in \mathcal{N}_h\right) \prod_{\substack{0 \le h' < h \\ h' \equiv 0 (\mathrm{mod}\ 2)}} x_{h'}^{\mathbb{1}\left(\Delta(S_{h'}^h) < y_{h'}\right)}\right] = \mathbb{E}\left[\mathbb{1}\left(S^h \in \mathcal{N}_h\right)\right] \prod_{\substack{0 \le h' < h \\ h' \equiv 0 (\mathrm{mod}\ 2)}} \mathbb{E}\left[x_{h'}^{\mathbb{1}\left(\Delta(S_{h'}^h) < y_{h'}\right)}\right]$$

$$= \mathbb{P}\left[S^h \in \mathcal{N}_h\right] \prod_{\substack{0 \le h' < h \\ h' \equiv 0 (\mathrm{mod}\ 2)}} \left(1 + x_{h'} \mathbb{P}\left[\left(\Delta(S_{h'}^h) < y_{h'}\right)\right]\right)$$

$$\le \mathbb{P}\left[S^h \in \mathcal{N}_h\right] \prod_{\substack{0 \le h' < h \\ h' \equiv 0 (\mathrm{mod}\ 2)}} \left(1 + a x_{h'} y_{h'}^b\right)$$

$$\le \mathbb{P}\left[S^h \in \mathcal{N}_h\right] \exp\left(\sum_{0 \le h' < h, h' \equiv 0 (\mathrm{mod}\ 2)} a x_{h'} y_{h'}^b\right)$$

$$\le \mathbb{P}\left[S^h \in \mathcal{N}_h\right] \exp\left(\sum_{0 \le h' < h, h' \equiv 0 (\mathrm{mod}\ 2)} \xi_2 \gamma^{(b-2)h'}\right)$$

$$\le \mathbb{P}\left[S^h \in \mathcal{N}_h\right] \exp\left(\sum_{h'=0}^{\infty} \xi_2 \gamma^{(b-2)h'}\right)$$

$$\le \mathbb{P}\left[S^h \in \mathcal{N}_h\right] \xi_3$$

We then conclude with

$$\mathbb{P}\left[S^h \in \mathcal{N}_h\right] \le \prod_{\substack{0 \le h' < h \\ h' \equiv 0 (\mathrm{mod}\ 2)}} a y_{h'}^b$$

$$\le \prod_{\substack{0 \le h' < h \\ h' \equiv 0 (\mathrm{mod}\ 2)}} a \left(16 \frac{\gamma^{(h-h')/2}}{\gamma(1-\gamma)}\right)^b$$

$$= (1/K^{h/2}) \prod_{\substack{0 \le h' < h \\ h' \equiv 0 (\mathrm{mod}\ 2)}} Ka \left(16 \frac{\gamma^{(h-h')/2}}{\gamma(1-\gamma)}\right)^b$$

$$\le \xi_4 / K^{h/2}.$$

$\square$

## C  On the choice of the near-optimality definition

In this part, we discuss the possible alternative choices to the near-optimality set $\mathcal{N}_h$ and explain the choice we made in Definition 1. From this definition, the knowledge of the rewards of the near-optimal nodes is enough to compute an optimal policy. Therefore, whatever the rewards associated with the non-near-optimal nodes are, they do not change the optimal policy. A good *adaptive* algorithm, would not exploring the whole tree, but would with high probability only explore a set not significantly larger than a set $\mathcal{N}_h$ of near-optimal nodes. There are other possible definitions of the near-optimality set satisfying this desired property that we could possibly consider for the definition of $\mathcal{N}_h$. Ideally, this set would be as small as possible.

**Alternative definition 1**  An idea could be to consider for any node $s$ and an ancestor $s'$, the value of the policy starting from $s'$ choosing the action leading to $s$ every time it can and playing optimally otherwise. Let $\mathcal{V}_s(s')$ be this value. A natural approach would be to define $\mathcal{N}_h$ only based on $\mathcal{V}_s(s_0)$

with $s_0$ being the root. To ensure that exploring the near-optimal nodes is enough to compute the optimal policy we need any near-optimal nodes $s$ of depth $h$ to verify

$$\left(\mathcal{V}\left[s_0\right] - \mathcal{V}_s(s_0)\right) p(s|s_0) \leq \frac{\gamma^h}{1-\gamma}.$$

When there is no `AVG` node, $p(s) = 1$ and this definition coincides with the near-optimal set defined in `OLOP` [5]. Nevertheless, in our case, $p(s)$ can be low and even 0 in the infinite case. When $p(s|s_0) = 0$ for all $s$, any node would be near-optimal which is bad because then the near-optimality set is large and the sample complexity of the resulting algorithm suffers.

**Alternative definition 2**    There is a smarter way to define a near-optimality set. We could define it as all the nodes $s$ of depth $h$ such that

$$\forall h' \leq h, \quad \sum_{i=h'}^{h} p(s_i|s_{h'})\gamma^i \left(\mathcal{V}\left[s_i\right] - \mathcal{V}\left[c(s_i, s)\right]\right) \leq \frac{\gamma^h}{1-\gamma}.$$

Taking $h' = 0$, we recover the alternative definition 1. Alternative definition 2 defines a smaller near-optimality set, as the condition needs to be verified for all $h'$ and not just for $h' = 0$. This definition would also lead to a smaller near-optimality set than our Definition 1. Indeed, in Definition 1 we consider only the first term of the sum above. This enables the algorithm to compute its strategy from a node $s$ only based on the value of its descendants. This is an ingredient leading to better computational efficiency of `TrailBlazer`. Indeed, the computational complexity of our algorithm is linear in the number of calls to the generative model. In addition, when the transition probabilities are low, both definitions are close and coincide in the infinite case, when the probability transitions are continuous and therefore $p(s|s') = 0$. In this case ($p(s|s') = 0$), for any set of nodes of not null probability mass, it is possible to change the value of the rewards on this subet such that the value of the root change. In this sense, this definition of near-optimal nodes is minimal when $p(s|s') = 0$.

## D  Auxiliary material

**Lemma 6.** *If for some $\alpha > 0, \beta, M$, a random variable $X$ verifies that*

$$\mathbb{E}\left[e^{\lambda X}\right] \leq M \exp\left(|\lambda|\alpha + \frac{\beta^2 \lambda^2}{2}\right),$$

*then for all $u > 0$*

$$\mathbb{P}\left[|X| \geq u\right] \leq 2M \exp\left(-\frac{(u-\alpha)^2}{2\beta^2}\right).$$

*Proof.*  Using Markov inequality for $u > \alpha$ and setting $\lambda = \frac{u-\alpha}{\beta^2}$, we get

$$\mathbb{P}\left[X \geq u\right] \leq \frac{\mathbb{E}\left[e^{\lambda X}\right]}{e^{\lambda u}} \leq M \exp\left(|\lambda|\alpha - \lambda u + \frac{\beta^2 \lambda^2}{2}\right) \leq M \exp\left(-\frac{(u-\alpha)^2}{2\beta^2}\right).$$

Similarly, for $u < -\alpha$,

$$\mathbb{P}\left[X \leq u\right] \leq M \exp\left(-\frac{(u+\alpha)^2}{2\beta^2}\right).$$

Taking a union bound, we have that for all $u > \alpha$,

$$\mathbb{P}\left[|X| \geq u\right] \leq 2M \exp\left(-\frac{(u-\alpha)^2}{2\beta^2}\right).$$

$\square$

**Lemma 7.** *For any positive random variable $X$ and event $A$,*

$$\mathbb{E}\left[X|A\right] \leq \frac{\mathbb{E}\left[X\right]}{\mathbb{P}\left[A\right]}.$$

*Proof.*

$$\mathbb{E}\left[X\right] = \mathbb{E}\left[X|A\right]\mathbb{P}\left[A\right] + \mathbb{E}\left[X|A^c\right]\mathbb{P}\left[A^c\right] \geq \mathbb{E}\left[X|A\right]\mathbb{P}\left[A\right]$$

$\square$