[Reviews · NeurIPS 2016]

Reviewer 1

Summary

This paper introduces a new Monte Carlo-based planning algorithm called TrailBlazer. The algorithm assumes the access to the generative model of the MDP, and its goal is to find the optimal value function of the root node V(s0) with as few call to the generative model as possible. The setting is discounted reward. The algorithm works for both finite and infinite state spaces. The paper provides several theoretical guarantees: PAC consistency, a high-probability upper bound on the number of calls to the generative model for finite state spaces, and an upper bound on the expected number of calls for infinite state spaces. Depending on the scenario, the paper either 1) improves the previous worst-case upper bounds (e.g., for finite state space and with stochastic dynamic) or 2) is the same as the previous results (deterministic dynamic; or without control — the same as Monte Carlo), or 3) provides new results (infinite state space). The TrailBlazer algorithm alternates between two types of nodes: Avg and Max. An Avg node computes the average value of its children, which are generated according to the transition probability. So it is essentially a Monte Carlo estimator. The parameter m controls the variance of this estimator. The Max node tries to find the value of the maximizing node among its children. It does so by eliminating the children that cannot be a child with maximum value (with high probability).

Qualitative Assessment

This is a good paper. It introduces a new algorithm, which has a potential to be used in many applications. The algorithm not only has the basic theoretical justification (consistency), but also comes with certain guarantees that are stronger than what is already known (for finite state space with generative model and stochastic system). The paper is generally well-written. But I think it can be revised to give more intuition as why the proposed algorithm works better than other approaches. The intuition is somehow missing. Also because of the induction on the tree, the proofs are not very straightforward. I don’t know if much can be done about it though. I have some comments/questions: - What is the main reason that the guarantee in Theorem 3 is only in expectation? - The algorithm does not exploit the possible regularities of the value function, e.g., its smoothness. Is it possible to benefit from them, for example, similar to the StoSO algorithm (Valko, Carpentier, Munos, “Stochastic Simultaneous Optimistic Optimization,” ICML 2013)? - The paper mentions that for non-vanishing action-gaps, the dimension d can be set to zero. More realistically, the action-gaps can follow a distribution over states (as introduced and analyzed by Farahmand, “Action-Gap Phenomenon in Reinforcement Learning,” NIPS 2011). What can be said about such a case? - Section 3.1 (Separate bias and variance) is not very clear. In particular, please expand on “In doing so, their algorithms compute … . However in our planning …”. - Typos (The line numbers refer to the Supplementary material): L81: worst —> worse L84: an near-optimal —> a near-optimal L220: “and an a term” L229: “an problem-dependent” L472: “i.d.d.” Appendix D: At several places, \Delta is written as Delta L504: “one need” —> “one needs” ******* Thank you for your response.

Confidence in this Review

2-Confident (read it all; understood it all reasonably well)


Reviewer 2

Summary

The paper provides a new sampling algorithm for planning in MDPs. The planning algorithm returns an estimate of the value function corresponding to the starting node of a loop-free MDP whose generative model is available to the planner (that is, a random next step can be sampled from any state). The sample complexity of the algorithm is analyzed theoretically. These results show achieve and/or improve previous results for MDPs with finite branching factors, and are also applicable when the number of possible next states are countably infinite.

Qualitative Assessment

The paper considers an interesting and important problem. The results can be interpreted as a natural combination of the planning algorithm of Busoniou and Munos (2012) with the sampling method of Kearns et al (1999). However, the paper introduces a few more tricks to make this idea work (e.g., balances confidence intervals and uncertainties at different parts of the planning tree). The presentation is quite nice and the authors try to give the intuition behind the choices in designing the algorithm. The clarity could be improved by noting that the MAX part of the algorithm is in fact action elimination for best arm identification (can't you use some of the existing results instead of reproving everything from scratch?). I would be also happy to see a more detailed discussion about the parameter choices k_l/(1-\eta)^2 and \eta max(U_l\epsilon). Also, please mention at the introduction of the algorithm that the semantics will be explained later (or change the order of discussion). On the negative side, the complexity measures are similarly hard to interpret as in previous results, although this might be inevitable. In line 45 it is mentioned that a planning tree is considered only for simplicity, and the algorithm can be extended to general MDPs with loops with merging states. I cannot see how you'd manage to actually propagate the sampling results in this case. In earlier algorithms, when you compute the optimal policy, this leads to significant complications (computationally), and I have failed to see why these would not apply here. So please either include such an explanation or remove the corresponding remark. Infinite N: note that this is countably infinite. Also, would it be possible to reduce the infinite N case to the finite one by showing that one can actually neglect successor states with sufficiently low probability (relative to epsilon) as those states will never be sampled and they have marginal contribution to the value function. Is the presented complexity measure d_H superior to what would follow this way? Minor: - l. 59: Mention that you want to obtain an _epsilon_-optimal policy. - There are several typos in the paper (e.g., choosing the correct article). - p. 3, footnote: define delta - l. 423: output->outputs

Confidence in this Review

2-Confident (read it all; understood it all reasonably well)


Reviewer 3

Summary

The paper provides a more refined algorithm, with analysis, for Monte-Carlo planning. The algorithm cleverly tries to merge Monte Carlo sampling (without action selection) with planning by ignoring other action branches when there is enough evidence to unambiguously decide the best action at a node. This allows the algorithm to explore only a subset of states reachable by following near-optimal policies. Analysis shows that the sample complexity depends on a measure of the quantity of near-optimal states. The paper also shows that the algorithm/analysis improves over best previous worst-case bounds under various conditions. The algorithm appears to be easy to implement.

Qualitative Assessment

With a single relatively simple algorithm, the authors achieve a number of things: improved worst case bound when the number of states N is finite, bounds that depends on the size of the space explored by near-optimal policy, conditions that allows polynomial sample complexity when N is infinite, and behaviour that is similar to Monte-Carlo sampling when there is a gap between the value of the best action compared to other actions. This is nice theoretical progress. Although the algorithm is relatively simple to implement, it appears to me that the progress is still mainly theoretical -- a very large tree needs to be explored before useful results can be obtained at the root, unlike practical algorithms that are mainly anytime algorithms. Can an anytime version be developed? Otherwise, the impact may be limited to theory. After author feedback: Thanks for the answer. Since the algorithm can be made anytime, it would be interesting to see experiments with it in the future.

Confidence in this Review

2-Confident (read it all; understood it all reasonably well)


Reviewer 4

Summary

The paper presents an algorithm to handle MDPs with a finite or infinite number of transitions from state-action to next states to return an estimate of the optimal value function at any state while minimizing the number of calls to the generative model, i.e., the sample complexity.

Qualitative Assessment

The major issue I have with this paper is that no experimental results on simulation domains or real applications have been reported. Althouth this is a theory paper, in my view, such results are required and should be reported to demonstrate the practicability and effiectiveness of the proposed algorithm, especially the authors themselve also claim that one of their main contributions is that their approach is easy to implement and computationally efficient. The text would benefit from further proof-reading. There are a number of typos: -Line 58: 1->[1] -Line 107: finite and infinite -> finite or infinite -Line 111: is be related->is related -Line 174: the concept of “opened” should be explained as it is the first time that it appears in the paper -Line 220: an a term -> a term -Line 355: using and approach->using an approach Figure 3 should be explained.

Confidence in this Review

1-Less confident (might not have understood significant parts)


Reviewer 5

Summary

This paper proposes a new algorithm called TrailBlazer for Monte-Carlo Planning. The authors provide the sample complexity of their algorithm. Their results include the case when there are infinite number of states.

Qualitative Assessment

This is a good paper with novel contribution. It propose a novel algorithm called TrailBlazer with good sampling complexity. The algorithm is also computational efficient. Instead of keeping an upper bound and lower bound of the value function, this paper creatively uses two parameters to control the error, one parameter for variance and another parameter for bias. Based on their algorithm, the authors prove the sample complexity. The claim that their framework has a polynomial sample complexity when there are infinite states is interesting. Also, their algorithm do need to visit all the policies. By this method, the algorithm becomes computationally efficient. To sum, the paper is solid and of good quality. Typos and suggestions: l.196 Grammar mistakes l.214 and l.205 The notation (\epsilon, m) is not consistent l.220 "and an a term" l.242 Need more explanation on why it behaves just like Monte-Carlo sampling.

Confidence in this Review

2-Confident (read it all; understood it all reasonably well)


Reviewer 6

Summary

Volunteer reviewer has no background in this research field, leave the comment as blank. Already contacted with the area chair.

Qualitative Assessment

Volunteer reviewer has no background in this research field, leave the comment as blank. Already contacted with the area chair.

Confidence in this Review

1-Less confident (might not have understood significant parts)